# Fine-Grained Uncertainty Quantification for Long-Form Language Model Outputs: A Comparative Study

**Dylan Bouchard**[*]     **Mohit Singh Chauhan**     **Viren Bajaj**     **David Skarbrevik**
*CVS Health, Wellesley, MA*

**Reviewed on OpenReview:** *https://openreview.net/forum?id=gngp4Zz9Sj*

## Abstract

Uncertainty quantification has emerged as an effective approach to closed-book hallucination detection for LLMs, but existing methods are largely designed for short-form outputs and do not generalize well to long-form generation. We introduce a taxonomy for fine-grained uncertainty quantification in long-form LLM outputs that distinguishes methods by design choices at three stages: response decomposition, unit-level scoring, and response-level aggregation. We formalize several families of consistency-based black-box scorers, providing generalizations and extensions of existing methods. We also introduce FactScore-STEM-Geo, a new 400-question long-form QA dataset spanning four categories across STEM and Geography. In our experiments across multiple LLMs and datasets, we find 1) claim-response entailment consistently performs better or on par with more complex claim-level scorers, 2) claim-level scoring generally yields better results than sentence-level scoring, and 3) uncertainty-aware decoding is highly effective for improving the factuality of long-form outputs. Our framework clarifies relationships between prior methods, enables apples-to-apples comparisons, and provides practical guidance for selecting components for fine-grained UQ.

## 1 Introduction

LLMs are widely used to generate long-form content for tasks such as summarization, open-ended question answering, and document drafting. These settings raise a basic reliability question: how confident should we be in a multi-sentence or multi-claim response, and where are the likely errors? Prior uncertainty quantification (UQ) methods for hallucination detection are mostly short-form (Huang et al., 2025; Tonmoy et al., 2024; Shorinwa et al., 2025; Huang et al., 2024a), which limits their ability to localize errors and effectively score confidence at the response level for long-form generation (Bakman et al., 2025; Vashurin et al., 2025b). In contrast to short-form methods, which estimate confidence for a single span, long-form methods decompose a response into granular units (sentences or claims) and score each unit. These scores can then be aggregated to produce a more precise response-level confidence estimate.

We study fine-grained uncertainty quantification for long-form LLM outputs, focusing on black-box methods that measure semantic consistency in sampled responses generated from the same prompt. Our methods follow a common pipeline: decompose a response into units (sentences or claims), score each unit using evidence derived from model behavior, and aggregate these scores into a response-level confidence. We introduce a taxonomy that distinguishes scoring methods by their design choices at the three stages of this process: how the response is decomposed, how unit-level scores are computed (matching scheme, semantic consistency measurement, and functional form), and how unit scores are aggregated. This taxonomy unifies, generalizes, and extends prior fine-grained UQ work and provides practical guidance for selecting components under different risk and deployment constraints.

Within this framework, we define several families of fine-grained scorers, each generalizing one or more existing methods. Unit-response scorers directly compare each unit in the original response with full-text

---

[*]Correspondence to `dbouchard92@gmail.com`

sampled responses using entailment signals (Zhang et al., 2024). Matched-unit scorers decompose both original and sampled responses, match each original unit to its most similar unit in every sample and then score consistency (Zhang et al., 2024). Graph-based scorers build an entailment graph over the union of unique claims across sampled responses and use graph centrality to estimate claim confidence (Jiang et al., 2024). Unit-QA scorers convert each claim or sentence to a question, sample LLM answers to those questions, and measure consistency among those answers to score the claim (Farquhar et al., 2024). We also decouple unit-level scoring from response-level aggregation and study aggregation operators that allow users to trade off factual precision and recall. This framework provides a common language for comparing methods, enables apples-to-apples evaluation of previous approaches, and clarifies how to assemble new scorers from interchangeable components.

We evaluate these fine-grained scoring methods on long-form hallucination detection across multiple LLMs and benchmarks, including FactScore-STEM-Geo, a new 400-question long-form QA benchmark we introduce spanning Chemistry, Physics, Biology, and Geography, comparing unit-level detection, calibration, and response-level scoring. To our knowledge, prior work compares subsets of these methods pairwise, whereas our study jointly evaluates a broad suite of claim-level and sentence-level scorers within a single framework. In our experiments, we find 1) claim-response entailment consistently performs better or on par with more complex claim-level scorers, 2) claim-level scoring generally yields better results than sentence-level scoring, and 3) uncertainty-aware decoding is highly effective for improving the factuality of long-form outputs. All methods introduced and evaluated in this work are available in our companion open-source toolkit, `uqlm` (Bouchard et al., 2026).[1]

## 2 Related Work

Short-form UQ typically compares an original response to sampled alternatives or inspects model-internal probabilities. For black-box UQ, exact-match style measures such as repetition and diversity quantify consistency across candidates but over-penalize paraphrases (Cole et al., 2023). Text-similarity metrics relax this by using n-gram metrics (ROUGE, BLEU, METEOR) (Lin, 2004; Papineni et al., 2002; Banerjee & Lavie, 2005; Shorinwa et al., 2025), sentence-embedding cosine similarity with Sentence-BERT (Reimers & Gurevych, 2019; Qurashi et al., 2020; Shorinwa et al., 2025), or BERTScore (Zhang* et al., 2020; Manakul et al., 2023). Natural Language Inference provides calibrated semantic signals via $P(\text{entailment})$ or $1 - P(\text{contradiction})$ between responses (Chen & Mueller, 2024; Lin et al., 2024), and has been used for semantic entropy and clustering-based variants (Kuhn et al., 2023; Kossen et al., 2024; Farquhar et al., 2024; Nikitin et al., 2024), as well as semantic density (Qiu & Miikkulainen, 2024). For white-box methods, token-probability aggregations include average and maximum negative log probability (Manakul et al., 2023), perplexity (Fadeeva et al., 2024), response improbability (Fadeeva et al., 2024), entropy over next-token distributions (Malinin & Gales, 2021; Manakul et al., 2023), and the geometric mean of token probabilities (Malinin & Gales, 2021).

Short-form UQ methods have been shown to generalize poorly to long-form LLM outputs (Bakman et al., 2025; Vashurin et al., 2025b). Fine-grained methods for long-form UQ address these limitations by first decomposing responses into granular units (sentences or claims) and then scoring each unit. LUQ measures whether sampled responses entail each original sentence, yielding sentence-level confidence scores, while LUQ-atomic applies this to extracted atomic claims to better target factual units (Zhang et al., 2024; 2025). LUQ-pair addresses NLI limitations on long texts by matching each original sentence to its most similar counterpart in sampled responses before computing entailment scores (Zhang et al., 2024). Jiang et al. (2024) obtain the union of unique claims over sampled responses, construct a bipartite graph of claim-response entailment, and measure confidence from graph centrality. Long-form semantic entropy decomposes a response into claims, generates questions whose answers are the claims given context, samples multiple answers, and computes semantic entropy across these answers (Farquhar et al., 2024). Other work focuses on claim-level calibration (Zhang et al., 2025; Huang et al., 2024b; Liu et al., 2024; Liu & Wu, 2025), and directly scoring entire long-form responses without a fine-grained approach (Yuan et al., 2025; Vashurin et al., 2025b; Da et al., 2024).

---

[1]https://github.com/cvs-health/uqlm

## 3 Fine-Grained Black-Box UQ Methods

### 3.1 Problem Setup and Taxonomy

We study fine-grained uncertainty quantification for long-form LLM outputs and formalize a taxonomy of methods aligned with a common three-stage pipeline. After response generation, all methods we consider (1) decompose a response into units (sentences or claims), (2) assign a unit-level confidence score to each unit given some form of evidence, and (3) aggregate unit scores into a response-level confidence. Methods differ in the design choices made at these three stages. We now introduce formal notation and summarize the corresponding design space.

Let $\mathcal{T}$ be the set of all finite text strings and let $\mathcal{Y} \subseteq \mathcal{T}$ denote the space of possible LLM responses. For a granularity $g \in \{\text{sent}, \text{claim}\}$, let $\mathcal{S}_g \subseteq \mathcal{T}$ denote the set of all possible textual units at granularity $g$, with $\mathcal{S}_{\text{sent}}$ containing all possible sentences and $\mathcal{S}_{\text{claim}}$ containing all possible atomic claims. Our goal is to produce unit-level uncertainty-aware confidence scores and to aggregate them into a response-level confidence for any $y \in \mathcal{Y}$. Scores lie in $[0,1]$ and may be thresholded at level $\tau \in [0,1]$ to flag likely hallucinations at either the unit or response level.

| Stage | Role (notation) | Main design choices |
|---|---|---|
| Decomposition | $\delta_g : \mathcal{Y} \to \mathcal{P}(\mathcal{S}_g)$ | Granularity $g \in \{\text{sent}, \text{claim}\}$ |
| Unit scoring | $c_g : \mathcal{S}_g \times \mathcal{E} \to [0,1]$ | Matching scheme; semantic consistency function; scorer functional form |
| Aggregation | $A : [0,1]^* \to [0,1]$ | Aggregation operator |

Table 1: Three-stage pipeline for fine-grained black-box scoring and associated design choices.

**Response decomposition.** We define a *decomposition function* $\delta_g : \mathcal{Y} \to \mathcal{P}(\mathcal{S}_g)$, that deconstructs a response $y \in \mathcal{Y}$ to a set of granular units, (i.e., sentences or claims), where granularity $g \in \{\text{sent}, \text{claim}\}$ and $\mathcal{P}(\cdot)$ is the power set operator. The main design choice at this stage is the granularity $g$. When $g = \text{sent}$, $\delta_g(\cdot)$ implements rule-based or model-based sentence segmentation. When $g = \text{claim}$, $\delta_g(\cdot)$ implements LLM-based claim extraction.[2]

**Unit-level confidence scoring.** A *unit-level confidence scorer* is a function $c_g : \mathcal{S}_g \times \mathcal{E} \to [0,1]$, that returns a confidence score for a unit $s \in \mathcal{S}_g$, where $\mathcal{E}$ denotes an evidence domain that will vary by scorer family. Scorers are defined such that higher values indicate greater likelihood of factual correctness. Given evidence $E \in \mathcal{E}$, a unit $s$ is flagged as possible hallucination if $c_g(s; E) < \tau_g$ for some threshold $\tau_g$. Design choices include (i) the *matching scheme* (e.g., unit-response or unit-unit), (ii) the *semantic consistency measurement* (e.g., NLI-based scores or embedding-based similarity; details in Section 3.2), and (iii) the *functional form* of $c_g$.

**Response-level aggregation.** Given unit scores $\{c_g(s; E)\}_{s \in \delta_g(y)}$ for a response $y$, an aggregation operator is a function $A : [0,1]^* \to [0,1]$ that maps an unspecified but finite number of unit scores to a single value in $[0,1]$. The resulting response-level confidence can then be thresholded at a threshold $\tau_y$ to flag likely hallucinations. Design choices here concern the functional form of $A$, for example simple averaging versus uncertainty-aware decoding with claim-level filtering. Detailed definitions are contained in Section 3.4.

### 3.2 Black-Box UQ and Semantic Consistency

Black-box UQ scorers exploit variation in LLM responses to the same prompt to assess semantic consistency. For a given prompt $x$, let $\mathbf{y}_{\text{cand}} = (y_1, ..., y_m)$ be $m$ candidate responses sampled via stochastic decoding (for example, non-zero temperature, top-$p$, or top-$k$), and let $y_0$ denote the response under audit, hereafter referred to as the *original response*.

---

[2]Importantly, the LLM used for decomposition need not be the same as the LLM used for response generation.

**Semantic consistency.** These methods rely on a *semantic consistency function* $\eta : \mathcal{T} \times \mathcal{T} \to [0, 1]$, to measure consistency between two strings of text. This consistency score typically measures either 1) semantic similarity (measured by embeddings) or 2) entailment or non-contradiction (estimated by natural language inference (NLI) model or an LLM). We consider the following semantic consistency functions in this work:

- **Entailment probability** - $p_e(a, b)$ is the NLI probability $b$ is entailed in $a$ (Lin et al., 2024).
- **Non-contradiction probability** - $1 - p_c(a, b)$ is the complement of NLI probability that $a$ is contradicted by $b$ (Chen & Mueller, 2024).
- **Contrasted entailment** - $\frac{p_e(a,b)}{p_e(a,b)+p_c(a,b)}$ is entailment probability excluding the *neutral* class (Zhang et al., 2024; Manakul et al., 2023).
- **Normalized cosine similarity** - $\hat{\cos}(a, b) = \frac{\cos(a,b)+1}{2}$ is the cosine similarity of $(a, b)$ normalized to $[0, 1]$ (Bouchard & Chauhan, 2025).
- **BERTScore** - $BertF1(a, b)$ is F1-BERTScore of $(a, b)$ computed with matched-token cosine similarity (Zhang* et al., 2020; Manakul et al., 2023).
- **Exact Match** - $\mathbb{I}[a = b]$ is an indicator function 1 if $a$ is identical to $b$ (Chen & Mueller, 2024).

Black-box UQ scores are then obtained by aggregating semantic consistency scores between the original response and sampled responses.

### 3.3 Fine-Grained Scorer Families

In contrast to traditional black-box UQ, where the semantic consistency function directly compares an original response with sampled responses, fine-grained adaptations perform comparisons on granular units obtained via the decomposition function $\delta_g$ with $g \in \{\text{sent}, \text{claim}\}$. We categorize fine-grained black-box UQ into four families of scorers: unit-response scorers, matched-unit scorers, unit-QA scorers, and graph-based scorers. Each family encompasses one or more existing methods as special cases and admits extensions through alternative semantic consistency functions and granularities. Table 2 summarizes the novelty status of each scorer configuration. Detailed definitions follow.

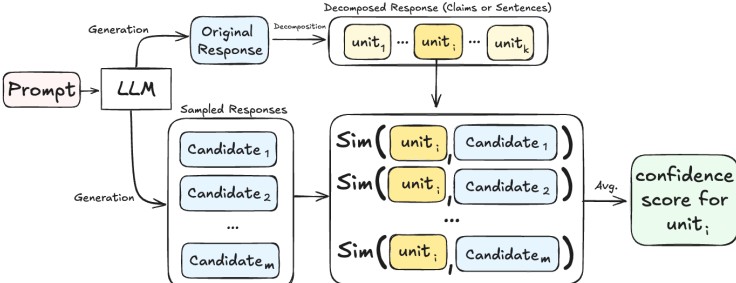

Figure 1: Unit-Response Scoring Workflow

### 3.3.1 Unit-Response Scorers

A *unit-response scorer* is a unit-level confidence scorer $c_g : \mathcal{S}_g \times \mathcal{Y}^m \to [0, 1]$ that compares individual sentences or claims contained in the decomposed original response $\delta_g(y_0)$ directly with candidate responses $\mathbf{y}_{\text{cand}} \in \mathcal{Y}^m$ (see Figure 1). Given a semantic consistency function $\eta(\cdot, \cdot)$, a unit-response scorer is specified as follows:

$$c_g(s; \mathbf{y}_{\text{cand}}) = \frac{1}{m} \sum_{j=1}^{m} \eta(y_j, s)$$

for $g \in \{\text{sent}, \text{claim}\}$ and $\eta \in \{p_e, 1 - p_c, \frac{p_e}{p_e+p_c}\}$.

For this family of scorers, we consider only NLI-based consistency, as the others defined above (normalized cosine similarity, BertScore, and exact match) are intended for texts of similar length and granularity, making them poorly suited for comparing granular units to full responses. Note that sentence-level and claim-level scoring with contrasted entailment are respectively equivalent to the LUQ and LUQ-atomic scorers proposed by Zhang et al. (2024). Our use of entailment and non-contradiction probability are straightforward extensions of LUQ and LUQ-atomic.

### 3.3.2 Matched-Unit Scorers

Rather than deconstructing only the original response, matched-unit scorers apply the decomposition function to all candidate responses as well. Specifically, a *matched-unit scorer*, given by $c_g : \mathcal{S}_g \times \mathcal{P}(\mathcal{S}_g)^m \to [0, 1]$, measures consistency between a unit (sentence or claim) from the original response and its most similar unit from each candidate response (Figure 2). Formally:

$$c_g(s; \mathbf{y}_{\text{cand}}) = \frac{1}{m} \sum_{j=1}^{m} \max_{s' \in \delta_g(y_j)} \eta(s', s)$$

for $\eta \in \{p_e, 1 - p_c, \frac{p_e}{p_e + p_c}, \hat{\cos}, BertF1\}$ and $g \in \{\text{sent}, \text{claim}\}$.

These matched consistency scores are averaged across pairings with all candidate responses. Note that scoring with $g = \text{sent}, \eta = \frac{p_e}{p_e + p_c}$ is equivalent to LUQ-pair (Zhang et al., 2024). We propose using alternative consistency functions $(p_e, 1 - p_c, \hat{\cos}, BertF1)$ as extensions of LUQ-pair.

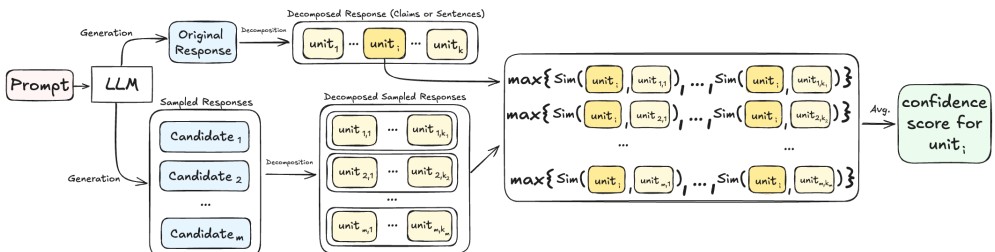

Figure 2: Matched-Unit Scoring Workflow

### 3.3.3 Unit-QA Scorers

The unit-QA family of scorers are inspired by the long-form version of semantic entropy proposed by Farquhar et al. (2024). We generalize this approach to allow for alternative consistency functions and granularities. The unit-QA approach relies on a *question inversion function* $\gamma : \mathcal{S}_g \to \mathcal{Y}$ that maps a unit (sentence or claim) to a question for which that unit is the answer. In practice, an LLM is used to create the question.[3]

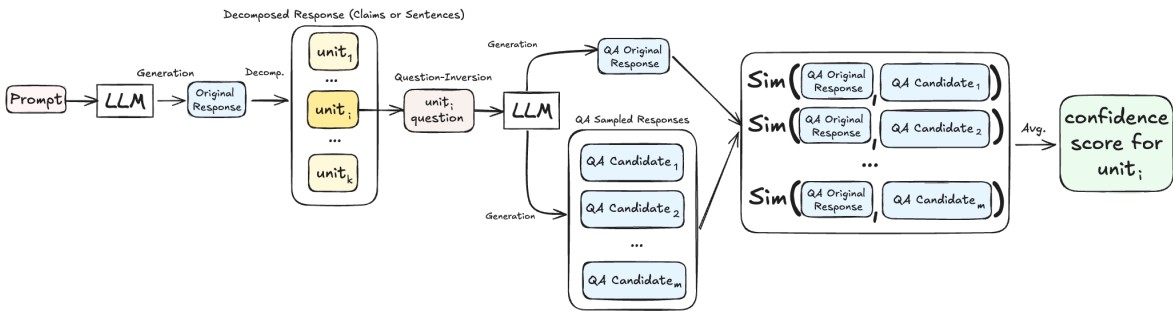

Figure 3: Unit-QA Scoring Workflow

---

[3]As with decomposition, this LLM need not be the same as the LLM used for response generation.

A *unit-QA scorer*, $c_g : \mathcal{S}_g \times \mathcal{Y}^{m+1} \to [0,1]$, measures consistency in multiple responses to unit question $\gamma(s)$:

$$c_g(s; y_0^{(s)}, \mathbf{y}_{\text{cand}}^{(s)}) = \frac{1}{m} \sum_{j=1}^{m} \eta(y_0^{(s)}, y_j^{(s)}),$$

where, for unit $s$, $y_j^{(s)}$ denotes the $j$th response to the unit's question $\gamma(s)$, $g \in \{\text{sent}, \text{claim}\}$, and $\eta \in \{1 - p_c, \text{côs}, BertF1, \mathbb{I}[y_0 = y_j]\}$.[4] Note that unit-QA effectively applies standard black-box UQ scoring to LLM responses to the unit questions (Figure 3). Hence, the semantic consistency functions we consider are standard black-box UQ choices as defined in Section 3.2.[5]

### 3.3.4 Graph-Based Scorers

Lastly, graph-based scorers, proposed by Jiang et al. (2024), decompose original and sampled responses into claims, obtain the union of unique claims across all responses, and compute graph centrality metrics (Hagberg et al., 2008) on the bipartite graph of claim-response entailment to measure uncertainty (Figure 4).[6] These scorers operate only at the claim level, as sentences typically contain multiple claims, meaning their union is not well-defined.

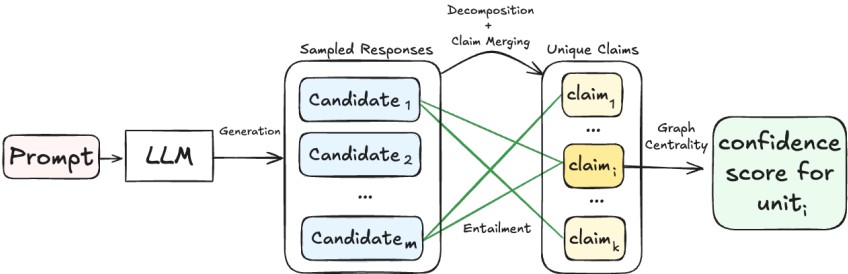

Figure 4: Graph-Based Scoring Workflow

Formally, we denote a bipartite graph $G$ with node set $V = \mathbf{s} \cup \mathbf{y}$, where $\mathbf{y}$ is a set of $m$ responses generated from the same prompt and $\mathbf{s}$ is the union of all unique claims across those decomposed responses. In particular, an edge exists between a claim-response pair $(s, y) \in \mathbf{s} \times \mathbf{y}$ if and only if claim $s$ is entailed in response $y$. We define the following graph metrics for claim $s$:

- **Betweenness Centrality** - $\frac{1}{B_{\max}} \sum_{u \neq v \neq s} \frac{\sigma_{uv}(s)}{\sigma_{uv}}$ measures uncertainty by calculating the proportion of shortest paths between node pairs that pass through node $s$, where $\sigma_{uv}$ represents all shortest paths between nodes $u$ and $v$, and $B_{\max}$ is the maximum possible value.[7]

- **Closeness Centrality** - $\frac{m + 2(|\mathbf{s}| - 1)}{\sum_{v \neq s} dist(s,v)}$ measures the inverse sum of distances to all other nodes, normalized by the minimum possible distance.

- **Harmonic Centrality** - $\frac{1}{H_{\max}} \sum_{v \neq s} \frac{1}{dist(s,v)}$ is the sum of inverse of distances to all other nodes, normalized by the maximum possible value, where $H_{\max} = m + \frac{|\mathbf{s}| - 1}{2}$.

- **Laplacian Centrality** - $\frac{E_L(G) - E_L(G_{-s})}{E_L(G)}$ is the proportional drop in Laplacian energy $E_L(G)$ resulting from dropping node $s$ from the graph, where $G_{-s}$ denotes the graph $G$ with node $s$ removed, $E_L(G) = \sum_i \lambda_i^2$, and $\lambda_i$ are the eigenvalues of $G$'s Laplacian matrix.

---

[4]Multiple questions may be generated per unit, with the unit score calculated as the average across questions.

[5]We also evaluate semantic entropy, which has a different functional form. It works by clustering mutually entailing responses, calculating cluster probabilities from response frequencies, and computing entropy across these clusters. For consistency with other confidence scorers, we convert this to semantic negentropy, following Bouchard & Chauhan (2025).

[6]The union of unique claims is obtained by sequentially merging claims from each sampled response using an LLM. As with decomposition and question-generation, this LLM need not be the same as the LLM used for response generation.

[7]Specifically, $B_{\max} = \frac{1}{2}[m^2(p+1)^2 + m(p+1)(2t-p-1) - t(2p-t+3)]$, $p = \frac{(|\mathbf{s}|-1)}{m}$, and $t = (|\mathbf{s}| - 1) \mod m$.

- **PageRank** - $\frac{1-d}{|V|} + d\sum_{v\in N(s)} \frac{C_{PR}(v)}{N(v)}$ is the stationary distribution probability of a random walk with restart probability $(1-d)$, where $N(s)$ denotes the set of neighboring nodes of $s$ and $C_{PR}(v)$ is PageRank of node $v$.

In contrast to Jiang et al. (2024), we exclude Eigenvector Centrality (preferring metrics with bounded support), while adding Laplacian and Harmonic Centrality. Note that we do not include Degree Centrality as a graph-based scorer as it is equivalent to claim-response entailment and does not require decomposing sampled responses.

| Family | Configuration | Status | Reference |
|---|---|---|---|
| Unit-Response | $g = \text{sent}, \eta = \frac{p_e}{p_e+p_c}$ | Equivalent | LUQ (Zhang et al., 2024) |
| | $g = \text{claim}, \eta = \frac{p_e}{p_e+p_c}$ | Equivalent | LUQ-atomic (Zhang et al., 2024) |
| | $g \in \{\text{sent}, \text{claim}\}, \eta = p_e$ | Generalization | Extension of LUQ / LUQ-atomic |
| | $g \in \{\text{sent}, \text{claim}\}, \eta = 1 - p_c$ | Generalization | Extension of LUQ / LUQ-atomic |
| Matched-Unit | $g = \text{sent}, \eta = \frac{p_e}{p_e+p_c}$ | Equivalent | LUQ-pair (Zhang et al., 2024) |
| | $g = \text{claim}, \eta = \frac{p_e}{p_e+p_c}$ | Generalization | Extension of LUQ-pair (granularity) |
| | $g \in \{\text{sent}, \text{claim}\}, \eta = p_e$ | Generalization | Extension of LUQ-pair |
| | $g \in \{\text{sent}, \text{claim}\}, \eta = 1 - p_c$ | Generalization | Extension of LUQ-pair |
| | $g \in \{\text{sent}, \text{claim}\}, \eta = BertF1$ | Generalization | Extension of LUQ-pair |
| | $g \in \{\text{sent}, \text{claim}\}, \eta = \hat{\cos}$ | Generalization | Extension of LUQ-pair |
| Unit-QA | $g = \text{claim}, \eta = \text{Semantic Entropy}$ | Equivalent$^\dagger$ | Long-form SE (Farquhar et al., 2024) |
| | $g = \text{sent}, \eta = \text{Semantic Entropy}$ | Generalization | Extension of Long-form SE (granularity) |
| | $g \in \{\text{sent}, \text{claim}\}, \eta = 1 - p_c$ | Generalization | Extension of Long-form SE |
| | $g \in \{\text{sent}, \text{claim}\}, \eta = BertF1$ | Generalization | Extension of Long-form SE |
| | $g \in \{\text{sent}, \text{claim}\}, \eta = \hat{\cos}$ | Generalization | Extension of Long-form SE |
| | $g \in \{\text{sent}, \text{claim}\}, \eta = \mathbb{I}[y_0 = y_j]$ | Generalization | Extension of Long-form SE |
| Graph-Based | Betweenness Centrality | Equivalent | Jiang et al. (2024) |
| | Closeness Centrality | Equivalent | Jiang et al. (2024) |
| | PageRank | Equivalent | Jiang et al. (2024) |
| | Harmonic Centrality | New | This work |
| | Laplacian Centrality | New | This work |

Table 2: Novelty classification of fine-grained scorers. We indicate whether each scorer configuration is equivalent to an existing method, a generalization of prior work (via alternative consistency functions or granularities), or newly proposed in this work. $^\dagger$Farquhar et al. (2024) use a less granular decomposition prompt and refer to these units as "factoids".

## 3.4 Response-Level Aggregation

We compute response-level confidence by averaging over the unit-level scores. Formally, for $g \in \{\text{sent}, \text{claim}\}$, this aggregation is given by:

$$A_{\text{avg}}(y) = \frac{1}{|\delta_g(y)|} \sum_{s\in\delta_g(y)} c_g(s; E).$$

While we treat averaging as our primary aggregation method, we additionally consider minimum, geometric mean, and rank-weighted average as alternatives.

**Uncertainty-aware decoding (UAD).** For claim-level scorers, we also implement an uncertainty-aware approach proposed by Jiang et al. (2024). This method first filters out low-confidence claims, then aggregates over the remaining ones. Specifically, we define the set of retained claims as

$$\widetilde{\delta}_{\text{UAD}}(y) = \{s \in \delta_{\text{claim}}(y) : c_{\text{claim}}(s; E) > \tau_{\text{claim}}\},$$

where $\tau_{\text{claim}}$ is a confidence threshold. The UAD confidence is then calculated by averaging over only these retained claims. This approach effectively modifies the response by retaining only high-confidence claims and using an LLM to reconstruct the response from the retained set.

## 4 Experiments

### 4.1 Experimental Setup

We evaluate fine-grained black-box UQ for hallucination detection on two long-form QA datasets. The first is **FactScore-bio** (Min et al., 2023), containing 500 prompts that instruct the model to write a biography about a given person. The second is **FactScore-STEM-Geo**, our new 400-question dataset spanning four diverse categories across STEM and Geography: Chemical Elements, Scientific Laws, Nerves in the human body, and Mountains.[8] For each of these four categories, we select the 100 entities with the longest Wikipedia articles and construct prompts that instruct the model to write some facts about the target topic. We generate an original response and 10 sampled responses for each question using five LLMs: Gemini-2.5-Flash, Gemini-2.5-Pro, GPT-4o, GPT-4o-mini, and Llama-4-Maverick-17B.

Each response is decomposed at two granularities: sentences using SpaCy (Honnibal et al., 2020) and claims using an LLM (Zhang et al., 2025). We compute claim-level confidence scores using claim-response, claim-QA (one question per claim), and graph-based scorers.[9] For sentence-level scores, we use sentence-response, matched-sentence, and sentence-QA (two questions per sentence) scorers. For unit-QA scoring, we generate an original response and 5 sampled responses per unit question. As an additional baseline, we compute unit-level verbalized confidence (Tian et al., 2023). For each family, scores are computed using applicable consistency functions.[10]

We obtain ground-truth factuality labels using the FactScore grading protocol, which uses an LLM to compare each unit in the response to the corresponding Wikipedia article text. For claim-level evaluation, we follow Jiang et al. (2024) by further classifying claims as objective or subjective and retaining only objective claims for evaluation, since subjective ones cannot be definitively verified.[11] We use Gemini-2.5-Flash for claim decomposition, claim merging, unit question generation, and grading as it offers robust performance balanced with computational efficiency and cost-effectiveness. Prompt templates are contained in Appendix D. We report per-response sentence and claim counts in Table 22 and LLM accuracies (the proportion of units graded as factually correct per the FactScore procedure) in Table 3.

| | FactScore-Bio | | | | | FactScore-STEM-Geo | | | | |
| Scorer | Gem-2.5-Fl | Gem-2.5-Pro | GPT-4o-Mini | GPT-4o | Llama-4 | Gem-2.5-Fl | Gem-2.5-Pro | GPT-4o-Mini | GPT-4o | Llama-4 |
|---|---|---|---|---|---|---|---|---|---|---|
| **Claim-Level Scorers** | | | | | | | | | | |
| Claim-response | 0.794 | 0.755 | 0.774 | 0.724 | 0.791 | 0.671 | 0.669 | **0.718** | 0.703 | 0.672 |
| Graph-based | **0.800** | **0.759** | **0.783** | **0.727** | **0.794** | **0.673** | **0.670** | 0.716 | **0.704** | **0.664** |
| Claim QA | 0.654 | 0.644 | 0.623 | 0.618 | 0.634 | 0.590 | 0.557 | 0.582 | 0.555 | 0.578 |
| LLM Accuracy | 0.723 | 0.733 | 0.647 | 0.792 | 0.686 | 0.707 | 0.685 | 0.680 | 0.741 | 0.734 |
| **Sentence-Level Scorers** | | | | | | | | | | |
| Sent-response | **0.678** | **0.694** | **0.695** | **0.652** | **0.716** | 0.590 | 0.565 | **0.670** | 0.632 | **0.626** |
| Mat-sentence | 0.632 | 0.661 | 0.620 | 0.615 | 0.696 | **0.596** | **0.573** | 0.669 | **0.633** | 0.606 |
| Sentence QA | 0.678 | 0.669 | 0.690 | 0.646 | 0.667 | 0.581 | 0.549 | 0.633 | 0.593 | 0.573 |
| LLM Accuracy | 0.462 | 0.455 | 0.347 | 0.532 | 0.411 | 0.500 | 0.491 | 0.487 | 0.584 | 0.541 |

Table 3: Unit-level AUROC (Higher is Better): Top Per Scorer Family by LLM and Dataset

---

[8] See Appendix C for code to recreate this dataset.

[9] We do not compute matched-claim scores, as their semantic comparison costs were prohibitive at our sample sizes. With $N_{\text{claim}} \approx 25$ claims per response and $m = 10$ sampled responses, matched-claim scoring requires approximately $m \cdot N_{\text{claim}}^2 = 6{,}250$ NLI comparisons per prompt, roughly $25\times$ the cost of matched-sentence scoring.

[10] NLI scores use `microsoft/deberta-large-mnli`.

[11] For sentence-level evaluation, we include all sentences regardless of objectivity classification, as sentences typically contain multiple claims of varying objectivity.

## 4.2 Unit-Level Classification and Calibration

We evaluate claim-level and sentence-level hallucination detection across all LLM-dataset combinations using FactScore grades as labels. Specifically, we assess classification performance with unit-level AUROC and AUPRC, and calibration with Brier Score (BS) and Expected Calibration Error (ECE). Comprehensive results are reported in Tables 11–18.[12]

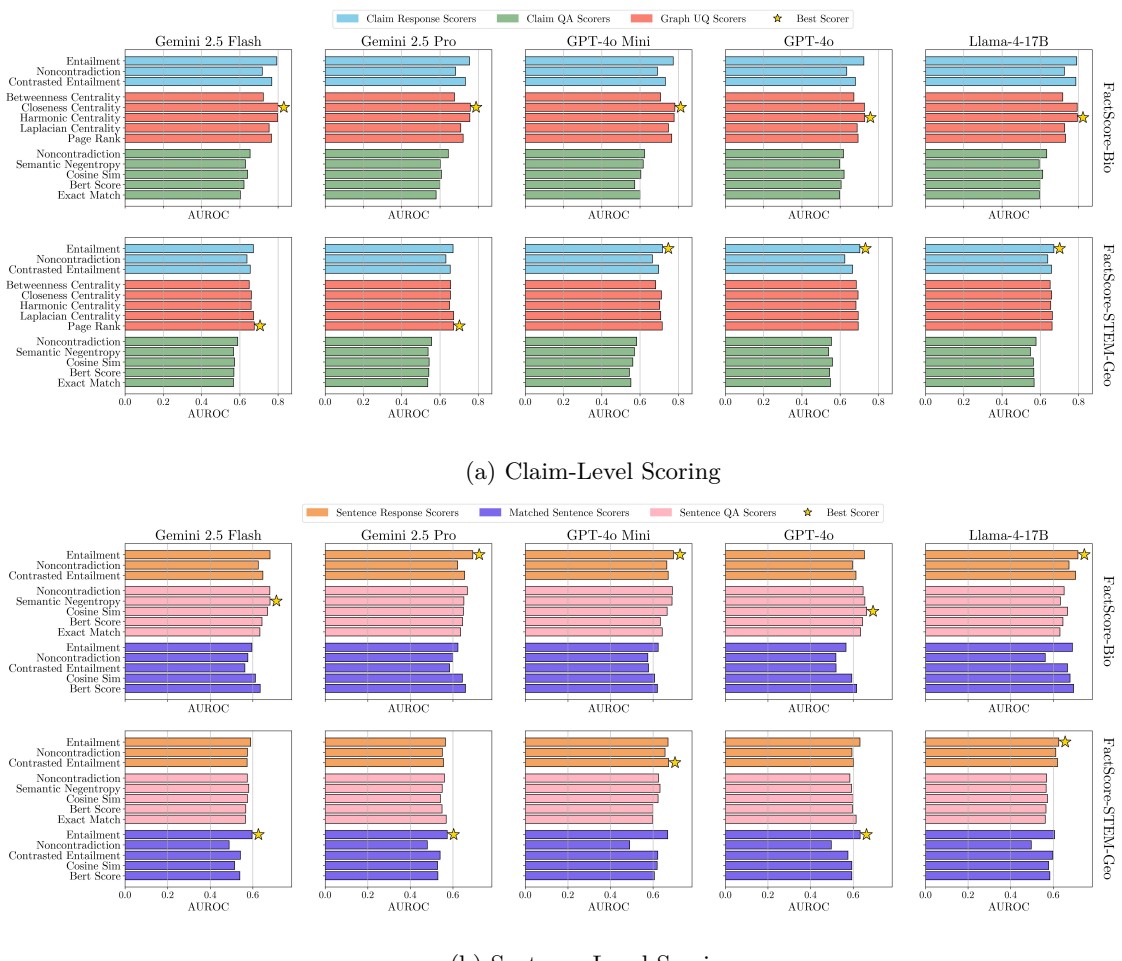

(a) Claim-Level Scoring

(b) Sentence-Level Scoring

Figure 5: Claim-Level and Sentence-Level AUROC by LLM and Dataset (Higher is Better)

**Classification.** Figure 5 shows claim-level and sentence-level AUROC, and Table 3 summarizes the top AUROC per family. For claim-level hallucination detection, scenario-specific top-scorer AUROC ranging from 0.67 for Gemini-2.5-Pro responses on FactScore-STEM-Geo (PageRank) to 0.80 for Gemini-2.5-Flash responses on FactScore-Bio (Closeness Centrality), on par with previous studies Zhang et al. (2025); Vashurin et al. (2025a); Farquhar et al. (2024). Comparing across scorer families, claim-response and graph-based scorers substantially outperform claim-QA approaches. Across the ten LLM-dataset combinations, claim-response entailment achieves the highest AUROC in 3 scenarios and is within 0.01 of the best scorer in the remaining 7, while also attaining the highest AUPRC in all 10. The top AUROC in the other 7 scenarios is achieved by graph-based scorers, with Closeness Centrality, PageRank, and Harmonic Centrality leading in 3, 2, and 1 scenarios, respectively. Among semantic consistency functions for claim-response scoring, entailment outperforms both contrasted entailment (used by Zhang et al. (2024)) and non-contradiction

---

[12]We also evaluated results category-wise for FactScore-STEM-Geo but found little variation across subsets.

across all ten scenarios. Among graph-based scorers, Page Rank and Closeness Centrality lead most often, each achieving highest AUROC in three scenarios, with Laplacian and Harmonic Centrality each leading in two.[13] Claim-QA scorers perform poorly overall, with non-contradiction being the top-performing claim-QA consistency function in 8 of 10 scenarios, yet rarely achieving AUROC above 0.6.[14] Verbalized confidence lags behind graph-based and claim-response but outperforms claim-QA scorers. AUPRC results exhibit similar patterns (Figure 13).

Sentence-level hallucination detection exhibits more varied performance across methods. The top-performing approaches are distributed across different scorer families, with matched-sentence, sentence-response, and sentence-QA scorers each achieving the highest AUROC in 5, 3, and 2 scenarios out of 10, respectively. Despite this variation, the maximum AUROC values across different scorer families remain similar within most scenarios. Consistent with Zhang et al. (2024), our results demonstrate that hallucination detection is inherently more difficult at the sentence level than at the claim level, as the highest sentence-level AUROC across all scenarios reaches only 0.716 and AUROC is consistently lower than claim-level scorers for the same datasets. For unit-response and matched-sentence scorers, our ablation studies (Appendix A.4) show that performance increases with the number of sampled responses but with substantial diminishing returns, with negligible gains beyond $m = 5$, consistent with previous work (Zhang et al., 2024; Jiang et al., 2024).

**Calibration.** Turning to calibration, we observe clear differences across families and granularities. At the claim level, most scorers are at best moderately calibrated and rarely attain ECE < 0.1. PageRank, Laplacian Centrality, and Betweenness Centrality are exceptions with notably worse calibration, with ECE values consistently above 0.6. Overall, our claim-level calibration metrics are comparable to those found in Zhang et al. (2025). At the sentence level, Sentence-QA Exact Match yields the lowest ECE in 8 of 10 scenarios. Apart from PageRank, Laplacian Centrality, and Betweenness Centrality, sentence-level scorers tend to be less calibrated overall than claim-level scorers. See Figure 14 for calibration plots of the best-calibrated scorers for FactScore-Bio.

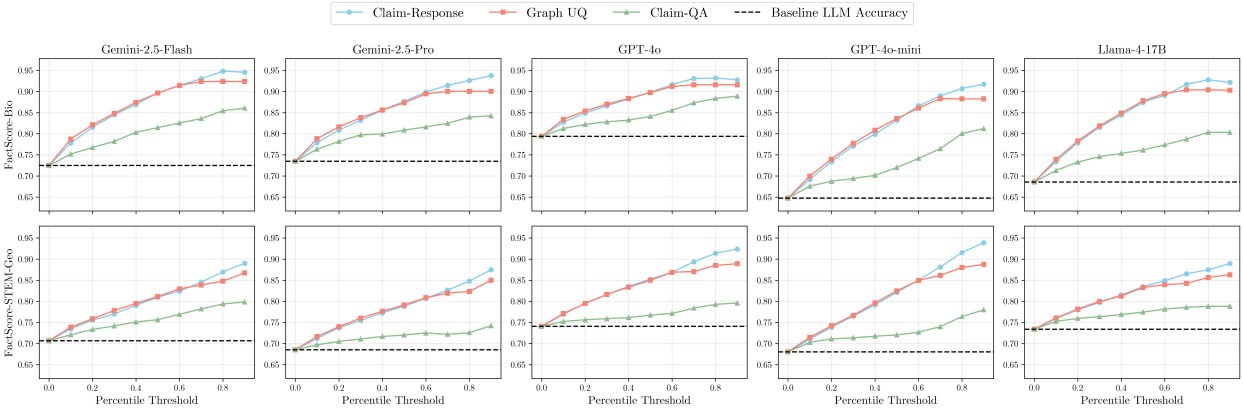

Figure 6: LLM Accuracy vs. UAD Filtering Threshold by LLM-Dataset (Top per Scorer Type)

### 4.3 Uncertainty-Aware Decoding

To evaluate the accuracy gains resulting from the UAD method proposed by Jiang et al. (2024), we analyze the relationship between LLM accuracy and claim-level filtering thresholds. We apply percentile-based thresholds at intervals of 10% (from the 10th to 90th percentile) for the top-performing scorer (by AUROC) from each scorer family. Figure 6 presents these results across all ten LLM-dataset combinations, with baseline LLM accuracy shown as a reference.

---

[13]Our results are generally similar to Jiang et al. (2024), with the notable exception that Closeness Centrality consistently outperforms other graph-based scorers in their experiments, which contain different LLMs and datasets.

[14]Farquhar et al. (2024) decompose responses into factoids containing several atomic claims and achieve better results. Manual inspection revealed many atomic claims are poorly suited for question-inversion; we discuss further in Section 5.

Our results demonstrate substantial accuracy improvements through uncertainty-aware filtering. For instance, when filtering claims from Gemini-2.5-Flash responses on FactScore-Bio at the 50th percentile threshold, accuracy increases dramatically from a baseline of 0.72 to approximately 0.90. This pattern is consistent across all evaluated scenarios, though with varying magnitudes of improvement. Specifically, claim-response-based filtering generally yields the most pronounced accuracy gains, followed closely by graph-based filtering, while claim-QA-based filtering, though still effective, shows more modest improvements. These findings highlight the practical utility of UAD across scoring methods for enhancing the factual reliability of long-form LLM outputs.

| Scorer | Gemini-2.5-Flash | Gemini-2.5-Pro | GPT-4o-Mini | GPT-4o | Llama-4-17B |
|---|---|---|---|---|---|
| **Claim-Level Scorers** | | | | | |
| CR Entailment | 0.73 | 0.69 | 0.68 | 0.56 | **0.70** |
| CR Non-Contradiction | 0.54 | 0.56 | 0.37 | 0.36 | 0.50 |
| CR Contrasted Entailment | 0.68 | 0.66 | 0.50 | 0.40 | 0.68 |
| QA Exact Match | 0.39 | 0.32 | 0.57 | 0.45 | 0.29 |
| QA Semantic Negentropy | 0.56 | 0.55 | 0.54 | 0.46 | 0.37 |
| QA Non-Contradiction | 0.55 | 0.61 | 0.36 | 0.46 | 0.55 |
| QA BERTScore | 0.47 | 0.49 | 0.41 | 0.48 | 0.34 |
| QA Cosine Similarity | 0.53 | 0.53 | 0.44 | 0.52 | 0.44 |
| Betweenness | -0.03 | -0.24 | -0.05 | -0.17 | -0.34 |
| Closeness | 0.74 | **0.71** | **0.70** | 0.57 | 0.67 |
| Harmonic | **0.74** | 0.70 | 0.68 | **0.60** | 0.66 |
| Page Rank | 0.51 | 0.47 | 0.60 | 0.24 | 0.05 |
| Laplacian | 0.49 | 0.44 | 0.57 | 0.25 | 0.12 |
| **Sentence-Level Scorers** | | | | | |
| SR Entailment | 0.47 | 0.48 | 0.50 | 0.25 | 0.36 |
| SR Non-Contradiction | 0.49 | 0.51 | 0.34 | 0.43 | 0.37 |
| SR Contrasted Entailment | 0.51 | 0.52 | 0.39 | 0.33 | 0.34 |
| Matched Entailment | 0.39 | 0.36 | 0.49 | 0.13 | 0.25 |
| Matched Non-Contradiction | 0.39 | 0.28 | 0.20 | 0.39 | 0.29 |
| Matched Contrasted Entailment | 0.39 | 0.44 | 0.27 | 0.10 | 0.19 |
| Matched Cosine Similarity | 0.50 | 0.50 | 0.49 | 0.30 | 0.40 |
| Matched BERTScore | 0.53 | 0.47 | 0.50 | 0.34 | 0.42 |
| QA Exact Match | 0.39 | 0.35 | 0.54 | 0.43 | 0.38 |
| QA Semantic Negentropy | 0.57 | 0.50 | 0.53 | 0.44 | 0.37 |
| QA Non-Contradiction | 0.53 | 0.53 | 0.46 | 0.34 | 0.40 |
| QA BERTScore | 0.47 | 0.47 | 0.48 | 0.48 | 0.44 |
| QA Cosine Similarity | 0.56 | 0.51 | 0.55 | 0.53 | 0.49 |
| **Short-Form Scorers** | | | | | |
| Semantic Negentropy | 0.47 | 0.35 | 0.39 | 0.31 | 0.31 |
| Non-Contradiction | 0.44 | 0.43 | 0.18 | 0.32 | 0.45 |
| BERTScore | 0.59 | 0.54 | 0.67 | 0.55 | 0.58 |
| Cosine Similarity | 0.56 | 0.55 | 0.54 | 0.37 | 0.47 |
| Normalized Sequence Probability | 0.03 | 0.03 | 0.56 | 0.47 | – |

Table 4: Pearson correlation coefficients (higher is better) between response-level confidence scores and factuality on FactScore-Bio by LLM.

### 4.4 Response-Level Scoring

Lastly, we average unit-level confidence scores to get response-level confidence scores and unit-level labels (FactScore grades) to get a response-level grade. For benchmarking purposes, we also compute confidence scores using four short-form black-box UQ scorers (Section 3.2): BERTScore (Manakul et al., 2023), cosine similarity (Shorinwa et al., 2025), non-contradiction (Lin et al., 2024), and semantic negentropy (Kuhn

et al., 2023; Bouchard & Chauhan, 2025), as well as one short-form white-box scorer: normalized sequence probability (Malinin & Gales, 2021).[15] We report Pearson and Spearman correlations between response-level scores and grades for both datasets in Tables 4, 19, 20, and 21. Pearson correlations are discussed below; Spearman correlations exhibit similar patterns for both datasets.

On FactScore-Bio (Table 4), the results are consistent with our unit-level evaluation. Closeness and Harmonic Centrality yield the strongest response-level signals, followed closely by claim-response entailment. These scorers outperform other fine-grained scorers as well as short-form baselines. Betweenness Centrality stands out as particularly ineffective, exhibiting no useful response-level signal. Sentence-level aggregation is noticeably weaker, with even the best sentence-level methods rarely approaching the correlations of the top claim-level scorers or the strongest short-form baselines. Among short-form UQ methods that directly score full responses, BERTScore is the most competitive, sometimes approaching the top claim-level scorers but generally falling short. Short-form white-box scoring is notably stronger for the GPT models compared to the Gemini models, with the latter providing negligible response level signal. Importantly, we note that even when competitive, short-form scorers do not enable localizing the uncertainty in the response or removing low-confidence claims via UAD.

On FactScore-STEM-Geo (Table 20), response-level scoring generally exhibits weaker performance relative to FactScore-Bio, consistent with the unit-level evaluation, with top correlations by model ranging from 0.37–0.62 compared to 0.60–0.74 on FactScore-Bio. Among aggregated claim-level scorers, QA non-contradiction and CR non-contradiction exhibit highest correlation, though considerably weaker than most claim-level scorers on FactScore-Bio. In contrast, the graph-based scorers that dominated FactScore-Bio are rarely competitive at the response level on this dataset. Betweenness Centrality again provides no useful signal. Sentence-level scorers are even weaker than on FactScore-Bio, with many cases of zero or negative correlation. Among short-form baselines, non-contradiction is especially strong for GPT-4o-Mini, achieving highest correlation out of all scorers for that model, an outlier relative to other models where short-form methods remain weaker. The shift in scorer rankings relative to FactScore-Bio may partly reflect compounding of noisier unit-level scores under aggregation.

While the results above use simple averaging to aggregate unit-level scores, we additionally evaluated alternative aggregation strategies: minimum (worst-case), geometric mean, and rank-weighted averaging. Geometric mean and rank-weighted averaging were competitive with simple averaging, achieving higher correlation in roughly one-fifth of LLM-scorer combinations, but simple averaging still performed best overall at the response level. Minimum aggregation performed notably worse, reducing Pearson correlations by 0.10–0.17 on average, suggesting isolated low-confidence claims are poor indicators of overall response factuality. Nevertheless, we note that minimum aggregation may be preferred in high-risk applications where flagging any potentially unsupported claim matters more than estimating average factuality.

## 5 Discussion

**Performance.** Our experiments in Section 4 highlight several consistent patterns on relative scorer performance. At the claim level, claim-response entailment generally exhibits the best performance, achieving the highest AUPRC in all scenarios and consistently performing better or on par with more complex graph-based and claim-QA methods. Graph-based scorers are competitive but rarely surpass claim-response entailment by a large margin, while claim-QA methods perform notably worse overall and often fail to reach AUROC values that are useful in practice. At the sentence level, hallucination detection is inherently more difficult, with top AUROC scores considerably lower than at the claim level. Performance varies across scorer families, but sentence-response entailment is a strong baseline, achieving the best AUROC in many scenarios and remaining competitive when matched-sentence or sentence-QA scorers perform slightly better.

Beyond detection, our uncertainty-aware decoding (UAD) experiments show that claim-level filtering can deliver large accuracy gains over baseline long-form generation. By discarding low-confidence claims and reconstructing responses from the remaining content, we observe consistent improvements across all evaluated

---

[15]Log-probabilities were not exposed by the inference API for Llama-4-17B during the evaluation phase for this specific model; consequently, results for this model do not include normalized sequence probability.

models and datasets, often boosting factual accuracy by a large margin even at moderate filtering thresholds. These gains are most pronounced when using claim-response and graph-based scorers.[16] Though these scorers are effective for ranking and filtering, their lack of calibration limits their utility for probability estimation.

**Unit-QA and Granularity.**  QA-based methods measure whether the primary LLM's responses to claim- or sentence-inverted questions are consistent. However, if a question admits multiple correct answers, inconsistency no longer signals factual error. The atomic nature of claim decomposition creates an inherent tension with this approach. Consider the claim "Katherine Ryan was an actor." The atomic granularity forces a dilemma: questions that preserve the claim's meaning either admit multiple correct answers or become trivially easy. For instance, "What was Katherine Ryan's profession?" fails because she holds multiple professions (comedian, writer, presenter, actress, singer), making "actor" not uniquely correct. Generalizing the question to "Who worked as an actor?" fails for the same reason, as thousands of individuals satisfy this criterion. Conversely, questions that would yield a unique answer, such as "What was Katherine Ryan's relation to the profession of acting?" or "Was Katherine Ryan an actor?", effectively encode the answer within the question itself, rendering verification meaningless. The superset sentence, "Katherine Ryan was a comedian, writer, presenter, actress and singer," resolves this tension by enabling well-formed questions like "What professions did Katherine Ryan hold?" where the complete list constitutes the unique correct answer. This helps explain why sentence-QA scorers outperform claim-QA methods in most cases, despite sentence-level classification being an inherently harder task.

**Computational Tradeoffs.**  We further quantify tradeoffs by analyzing both semantic consistency computations and LLM generation costs across scorer families (Table 5). Sentence-response and claim-response scorers decompose only the original response and compare each unit to $m$ sampled responses, yielding $mN_g$ semantic comparisons per response, where $N_g$ is the number of units at granularity $g$. Matched-sentence scorers decompose each of the $m$ sampled responses and, for every original sentence, compare against all sentences in each sample, yielding roughly $mN_{\text{sent}}^2$ semantic comparisons per prompt. Graph-based scorers decompose the original and all $m$ sampled responses into claims and construct an entailment graph over the union of $N_{m\text{-union}}$ unique claims across sampled responses, requiring approximately $mN_{m\text{-union}}$ consistency comparisons, where $N_{m\text{-union}} \approx 3N_{\text{claim}}$ on average in our experiments (Table 22). Beyond generating the sampled responses, graph-based methods also require decomposing each of the sampled responses into claims and sequentially merging claim sets, for a total of roughly $3m + 1$ additional generations per original response. Unit-QA scorers sample responses to unit questions rather than to the original prompt. Per unit, this requires a generation to obtain $Q$ questions and generating $m_{qa} + 1$ responses per question, yielding $((1 + m_{\text{qa}})Q + 1)N_g$ additional generations and $m_{\text{qa}}QN_g$ semantic comparisons per prompt. Relative to sentence-level scorers, all claim-level scorers require one additional generation for decomposing the original response and one additional generation if UAD is used.[17]

| Scorer Family | Comparisons | Additional Generations |
|---|---|---|
| Claim-Response | $mN_{\text{claim}}$ | $m + \mathbb{I}_{\text{UAD}} + 1$ |
| Graph-Based | $mN_{m\text{-union}}$ | $3m + \mathbb{I}_{\text{UAD}} + 1$ |
| Claim-QA | $m_{\text{qa}}QN_{\text{claim}}$ | $((1 + m_{\text{qa}})Q + 1)N_{claim} + \mathbb{I}_{\text{UAD}} + 1$ |
| Sentence-response | $mN_{\text{sent}}$ | $m$ |
| Matched-Sentence | $mN_{\text{sent}}^2$ | $m$ |
| Sentence-QA | $m_{\text{qa}}QN_{\text{sent}}$ | $((1 + m_{\text{qa}})Q + 1)N_{\text{sent}}$ |

Table 5: Per-prompt semantic consistency comparisons and LLM generations by scorer family.

Importantly, we note that the number of units at each granularity also differs substantially. In our experiments, $N_{\text{claim}} \approx 5N_{\text{sent}}$ (Table 22), so claim-level methods operate over several times as many units

---

[16]Graph-based scorers use all unique claims across samples, increasing content coverage in reconstructed responses.
[17]Sentence decomposition uses SpaCy instead of an LLM.

as sentence-level methods for the same prompt, amplifying generation and computation costs. Table 23 summarizes the semantic comparison counts and runtimes from our experiments.[18]

**Method Selection.** Taken together, our findings support the following practical recommendations. If computational resources permit claim-level analysis, claim-response entailment offers a robust default over more complex methods when both accuracy and efficiency are considered, with as few as 5 sampled responses typically being sufficient. When only sentence-level scoring is feasible, sentence-response entailment provides a simple, competitive, and cheap baseline. When claim-level scoring is used, for only one additional generation per response, uncertainty-aware decoding offers substantial accuracy improvements and turns fine-grained black-box UQ from a purely diagnostic tool into a mechanism for editing and improving long-form outputs.

**Broader Impact.** UAD produces responses that are more accurate but also more confident-seeming, which could increase over-reliance on outputs containing residual errors. This concern is amplified because low-confidence claims are filtered rather than flagged, reducing user visibility into model uncertainty. To mitigate this, our open-source implementation reports the original response, claim-level confidence scores for all claims, retention status for each claim, the refined UAD response, and pre-/post-UAD response-level confidence, ensuring the full uncertainty picture remains available to practitioners. Additionally, deploying UQ methods without proper validation in scientific or safety-sensitive domains carries real risk, as miscalibrated confidence scores could lead to unwarranted trust in erroneous outputs.

## 6 Conclusion

We introduced a general framework for fine-grained black-box uncertainty quantification in long-form LLM outputs and formalized a taxonomy aligned with a three-stage pipeline: response decomposition, unit-level confidence scoring, and response-level aggregation. Within this framework we define four scorer families and two aggregation strategies (simple averaging and uncertainty-aware decoding), unifying, generalizing, and extending existing fine-grained UQ methods. Experiments on five LLMs and two long-form QA datasets quantify performance, calibration, and computational trade-offs and yield practical recommendations: claim-response entailment is preferable when both performance and cost are considered, and claim-level uncertainty-aware decoding reliably improves long-form factual precision. To support reproducibility, all methods evaluated in this study are available in our accompanying open-source package, `uqlm`.

## 7 Limitations

We acknowledge several limitations of this work. First, due to the computational intensity of our experiments, comprehensively varying all experimental parameters was infeasible. In particular, we fix a single temperature, one NLI model, one embedding model for cosine similarity, and one LLM for claim decomposition and merging, question generation, and grading. Different choices for these components could change absolute and relative performance.[19] Similarly, we evaluate only two granularities (sentences and atomic claims); intermediate granularities such as factoid-level decomposition (Farquhar et al., 2024) may better balance the trade-offs between claim-QA compatibility and fine-grained scoring, and warrant future investigation. Second, while we evaluate five LLMs and two long-form QA datasets, our findings may not generalize to all models or domains. Differences in model behavior as well as dataset characteristics such as topic, difficulty, and answer format, may affect hallucination patterns and scorer effectiveness. Third, our experiments include two thinking models (Gemini-2.5-Pro and Gemini-2.5-Flash, which use chain-of-thought reasoning by default). While Gemini-2.5-Pro exhibits the lowest claim diversity ratio across both datasets (Table 22), this does not meaningfully degrade scorer performance. However, we do not systematically vary thinking budgets or evaluate dedicated reasoning models (e.g., o3, QwQ), which remains an interesting direction for future work. Fourth, our FactScore-STEM-Geo dataset selects entities with the longest Wikipedia articles per

---

[18]We do not report generation time, as it is heavily influenced by factors that are difficult to standardize, including deployment-specific rate limits and hardware configurations.

[19]We note that LLMs are subject to bias and variability.

category, biasing evaluation toward well-documented topics. Results may not generalize to less-documented entities, where hallucination risk is higher and UQ methods may be more needed. Finally, we focus solely on fine-grained, black-box, consistency-based methods, excluding white-box approaches (using token probabilities or attention) and long-form methods that score only at the response level, which may outperform our evaluated methods.

## Contribution Statement

**Research.**  Dylan Bouchard led the overall research effort, including literature review, taxonomy conceptualization and design, and experimental design.

**Writing.**  Dylan Bouchard wrote the introduction, related work, methods, discussion, conclusion, and limitations sections. Viren Bajaj created the figures used in the methods section. All authors contributed to the experiments section. Mohit Singh Chauhan provided manuscript edits throughout the writing process.

**Code and Experiments.**  The distribution of primary code and experiment contributions is detailed below.

- Unit decomposition: Viren Bajaj, David Skarbrevik, Dylan Bouchard
- Unit-response scoring: Viren Bajaj
- Matched-sentence scoring: Dylan Bouchard
- Unit-QA scoring: Mohit Singh Chauhan
- Graph-based scoring: David Skarbrevik
- Experiment implementation: Dylan Bouchard
- Dataset curation: David Skarbrevik, Dylan Bouchard
- FactScore grading: Viren Bajaj, Dylan Bouchard
- Descriptive analysis: Mohit Singh Chauhan
- Unit-level evaluation: Mohit Singh Chauhan
- Uncertainty-aware decoding evaluation: Dylan Bouchard
- Calibration evaluation: David Skarbrevik
- Response-level evaluation: Viren Bajaj
- Ablations and validations: Dylan Bouchard, Mohit Singh Chauhan

## Conflict of Interest

Mohit Singh Chauhan and Viren Bajaj are employed by, and receive stock and equity from, CVS Health Corporation. Dylan Bouchard and David Skarbrevik were employed by, and received stock and equity from, CVS Health Corporation at the time this work was conducted.

## Disclaimer

Prompts are included solely for reproducibility and do not imply endorsement or affiliation. Gemini is a trademark of Google and GPT is a trademark of OpenAI. This is an independent publication and has not been authorized, endorsed, or sponsored by Google or OpenAI.

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

# A  Additional Experiments

## A.1  Grader Robustness Analysis

A potential concern with our experimental setup is self-preference bias, as Gemini-2.5-Flash serves both as a subject LLM and as the grader for decomposition and correctness evaluation. To address this, we conducted a robustness check using GPT-4o as an alternative grader on a stratified sample of the data.

For four generator LLMs and both granularity levels (sentence and claim), we sampled 400 atomic units: 200 marked correct by Gemini-2.5-Flash and 200 marked incorrect. We then re-graded these samples using GPT-4o and computed inter-grader agreement metrics. Table 6 presents the agreement between Gemini-2.5-Flash and GPT-4o graders across all generator-granularity combinations.

| Generator | Granularity | Agreement (%) | % Correct (GPT-4o) | % Correct (Gemini) | Cohen's $\kappa$ |
|---|---|---|---|---|---|
| Gemini-2.5-Flash | Sentence | 81.5 | 44.0 | 50.0 | 0.63 |
| | Claim | 91.8 | 48.8 | 50.0 | 0.84 |
| GPT-4o | Sentence | 83.0 | 47.5 | 50.0 | 0.66 |
| | Claim | 92.0 | 48.0 | 50.0 | 0.84 |
| GPT-4o-mini | Sentence | 85.0 | 39.7 | 50.0 | 0.70 |
| | Claim | 90.3 | 43.8 | 50.0 | 0.80 |
| Gemini-2.5-Pro | Sentence | 82.9 | 44.7 | 50.0 | 0.66 |
| | Claim | 88.8 | 50.8 | 50.0 | 0.78 |

Table 6: Inter-grader agreement between Gemini-2.5-Flash and GPT-4o on a stratified sample (n=400 per condition). We report percent agreement, the proportion marked correct by each grader, and Cohen's $\kappa$.

The results demonstrate strong agreement between the two graders across all conditions. Cohen's $\kappa$ values range from 0.63 to 0.84, indicating substantial to almost perfect agreement. Notably, claim-level grading exhibits higher agreement ($\kappa = 0.78$–$0.84$) than sentence-level grading ($\kappa = 0.63$–$0.70$), likely because atomic claims are more precisely defined and easier to verify against reference documents, while sentences may contain both correct and incorrect claims.

Importantly, agreement levels are consistent regardless of whether the generator LLM matches the original grader (Gemini-Flash: $\kappa = 0.63, 0.84$) or not (GPT-4o: $\kappa = 0.66, 0.84$). This suggests that self-preference bias does not substantially affect our grading results. We do observe that GPT-4o tends to be slightly more conservative in marking units as correct (39.7%–50.8% vs. the stratified 50% from Gemini-Flash). However, this difference may reflect genuine performance disparities on long-context grading tasks rather than bias, as Gemini-2.5-Flash has been shown to outperform GPT-4o on long-context benchmarks (Bai et al., 2025). Regardless, this systematic difference does not undermine the relative comparisons central to our analysis.

## A.2  Claim Decomposition Validation

To validate the reliability of our LLM-based claim decomposition, two authors manually evaluated 410 claims across 15 responses (3 randomly sampled per dataset across FactScore-Bio and the four categories of FactScore-STEM-Geo). Each claim was assessed for: (1) faithfulness (whether the claim is factually correct relative to the source response); (2) standalone quality (whether the claim is interpretable without additional context, per the decomposition instructions), and (3) recall (whether any claims were missing from the decomposition).

Table 7 summarizes our findings. The decomposition achieves 99.8% faithfulness, with only 1/410 claims containing an objective factual error relative to the original response. An additional 13 claims (3.2%) were factually correct but required surrounding context to interpret, violating the standalone requirement of the decomposition prompt in Listing 5. This standalone requirement violation further highlights the difficulty of claim-QA methods, which require each atomic claim to be self-contained (e.g., avoiding pronouns such as "he," "she," or "it" and instead using the original subject; see Section 5 for further discussion). Recall was similarly high, with only 3 claims identified as missing across all 410 evaluated.

| Criterion | Result | Rate |
|---|---|---|
| Faithfulness | 409/410 | 99.8% |
| Standalone quality | 397/410 | 96.8% |
| Recall | 3 missing | ∼99.3% |

Table 7: Manual evaluation of claim decomposition quality.

Overall, these results demonstrate that the claim decomposition is highly reliable and suitable as a foundation for downstream claim-level verification and uncertainty quantification.

### A.3 Threshold Transfer Analysis

A practical consideration for deploying factuality scoring systems is whether decision thresholds learned on one dataset or scorer can transfer to new settings without additional labeled data. We investigate two transfer scenarios: (1) cross-dataset transfer, where a threshold tuned on one dataset is applied to another dataset using the same scorer, and (2) cross-scorer transfer, where a threshold tuned on one scorer is applied to a different scorer on the same dataset.

We conduct this analysis on Llama-4-Maverick-17B responses using the top scorer from each family (by AUROC), totaling 7,972 (Bio) and 9,842 (STEM-Geo) objective claims for claim-level analysis, and 2,701 (Bio) and 2,226 (STEM-Geo) sentences for sentence-level analysis. For each setting, we split the data 50/50 into calibration and test sets. On the calibration set, we find the threshold that maximizes F1 score by grid search over $[0, 1]$. We then evaluate this threshold on the test set, comparing self-tuned performance (threshold tuned on the same dataset or scorer) against cross-tuned performance (threshold transferred from a different dataset or scorer). For both cross-dataset and cross-scorer threshold transfer, we report both transfer directions separately.

Table 8 shows cross-dataset threshold transfer results. For claim-level scorers, thresholds transfer well across datasets, with gaps up to 0.028 (claim-response entailment on Bio). Claim-QA noncontradiction shows near-perfect transfer with essentially zero gap on both datasets. Sentence-level scorers exhibit larger gaps (up to 0.049 for sentence-response entailment on Bio), with QA-cosine similarity showing the best transfer (gaps of 0.007 and 0.000).

| Granularity | Scorer | Evaluate on Bio | | Evaluate on STEM-Geo | |
|---|---|---|---|---|---|
| | | Self | Cross | Self | Cross |
| Claim | Claim-response entailment | 0.840 | 0.812 | 0.844 | 0.829 |
| | Closeness centrality | 0.847 | 0.827 | 0.843 | 0.833 |
| | Claim-QA noncontradiction | 0.814 | 0.815 | 0.844 | 0.844 |
| Sentence | Sentence-response entailment | 0.625 | 0.576 | 0.699 | 0.681 |
| | Matched-sentence entailment | 0.601 | 0.576 | 0.699 | 0.666 |
| | QA-cosine similarity | 0.605 | 0.598 | 0.700 | 0.700 |

Table 8: Cross-dataset threshold transfer (same scorer, different dataset). Self: threshold optimized on target dataset. Cross: threshold transferred from other dataset. Small gaps indicate thresholds generalize well across datasets.

In contrast, Tables 9 and 10 show that cross-scorer threshold transfer performs substantially worse. Each cell shows the F1 score when using a threshold tuned on the row scorer to evaluate the column scorer. Diagonal entries (bold) represent self-tuned performance. For claim-level scorers, transfer is relatively robust, with most off-diagonal entries close to the diagonal. However, sentence-level scorers show severe degradation when transferring from QA-cosine similarity to entailment-based scorers (e.g., 0.403 and 0.312 on FactScore-Bio, 0.392 and 0.209 on FactScore-STEM-Geo). This result is not particularly surprising, as different scorers

produce different score distributions, meaning a threshold calibrated for one scorer may not be a useful decision boundary for another.

| | Tuned on | Evaluated on | | |
|---|---|---|---|---|
| | | CR Entailment | Closeness Cent. | Claim-QA Noncon. |
| FactScore-Bio | CR Entailment | **0.840** | 0.817 | 0.815 |
| | Closeness Cent. | 0.757 | **0.847** | 0.811 |
| | Claim-QA Noncon. | 0.839 | 0.817 | **0.814** |
| FactScore-STEM-Geo | CR Entailment | **0.844** | 0.844 | 0.844 |
| | Closeness Cent. | 0.756 | **0.843** | 0.843 |
| | Claim-QA Noncon. | 0.789 | 0.844 | **0.844** |

Table 9: Cross-scorer threshold transfer for claim-level scorers. Rows indicate which scorer's threshold is used; columns indicate which scorer is evaluated. Diagonal (bold) is self-tuned.

| | Tuned on | Evaluated on | | |
|---|---|---|---|---|
| | | SR Entailment | Matched-Sent. Entailment | QA-Cosine Sim. |
| FactScore-Bio | SR Entailment | **0.625** | 0.586 | 0.576 |
| | Matched-Sent. Entailment | 0.625 | **0.601** | 0.576 |
| | QA-Cosine Sim. | 0.403 | 0.312 | **0.605** |
| FactScore-STEM-Geo | SR Entailment | **0.699** | 0.699 | 0.699 |
| | Matched-Sent. | 0.699 | **0.699** | 0.699 |
| | QA-Cosine Sim. | 0.392 | 0.209 | **0.700** |

Table 10: Cross-scorer threshold transfer for sentence-level scorers. Rows indicate which scorer's threshold is used; columns indicate which scorer is evaluated. Diagonal (bold) is self-tuned. Transfer from QA-cosine similarity to entailment-based scorers fails dramatically.

Overall, these results suggest that practitioners can tune a threshold on a labeled sample from one domain and apply it to new domains with minimal performance loss, as long as the same scorer is used. However, switching to a different scorer requires re-tuning the threshold on labeled data.

### A.4 Sampled Response Ablations

Our ablation studies examine how the number of sampled responses $m$ affects performance for unit-response and matched-sentence scorers.[20]Figures 7–12 show that both AUROC and AUPRC generally increase monotonically with $m$ across datasets and models, but with substantial diminishing returns. For claim-response scorers, performance gains are most significant when increasing from $m = 1$ to $m = 3$, with approximate AUROC improvements of 0.05-0.07. Beyond $m = 5$, additional samples yield negligible improvements (typically <0.01 AUROC), consistent with findings from Zhang et al. (2024) and Jiang et al. (2024).

Similar patterns emerge for sentence-response and matched-sentence scorers, with some notable exceptions. In particular, we observe that gains from additional responses are negligible or non-existent in scenarios where hallucination detection performance is poor (i.e., AUROC values close to 0.5), such as with matched-sentence non-contradiction used on LLM responses to FactScore-STEM-Geo questions. Overall, these results indicate that practitioners can confidently limit sampling to just a few responses without sacrificing meaningful hallucination detection performance for these scorers. This finding is particularly useful in practice given the high computational and generation costs that scale with number of sampled responses.

---

[20]We did not conduct ablations for graph-based scorers due to the high computational costs of sequential claim merging, nor for claim-QA scorers given their relatively lower performance compared to other methods.

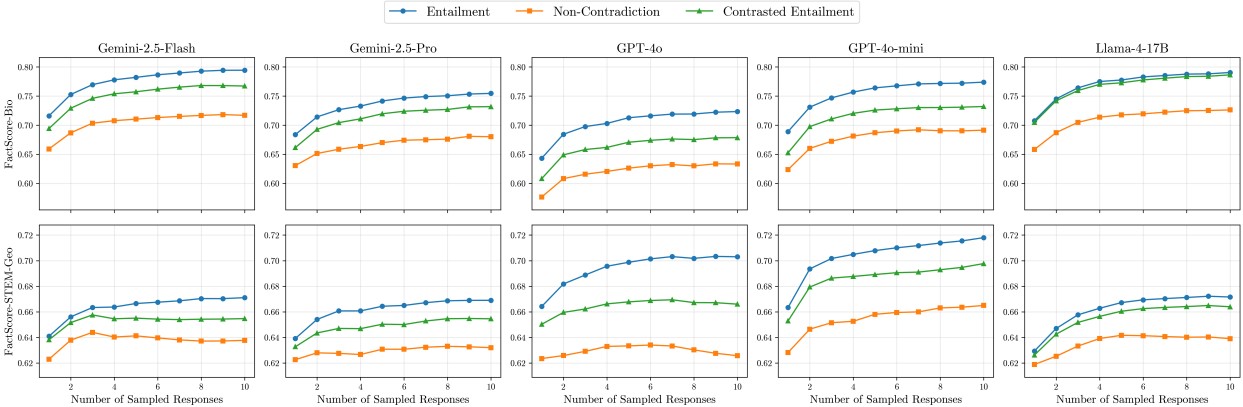

Figure 7: Claim-Response Hallucination Detection AUROC by Number of Sampled Responses

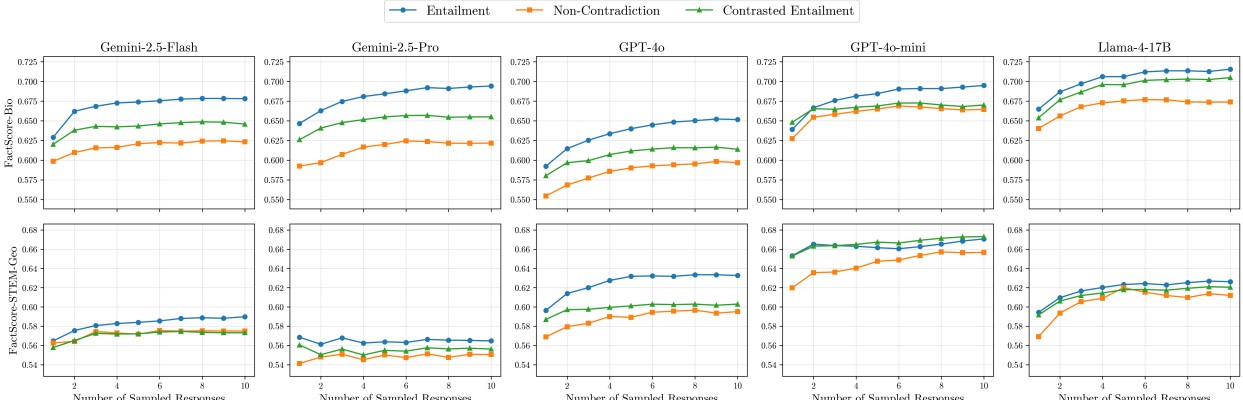

Figure 8: Sentence-Response Hallucination Detection AUROC by Number of Sampled Responses

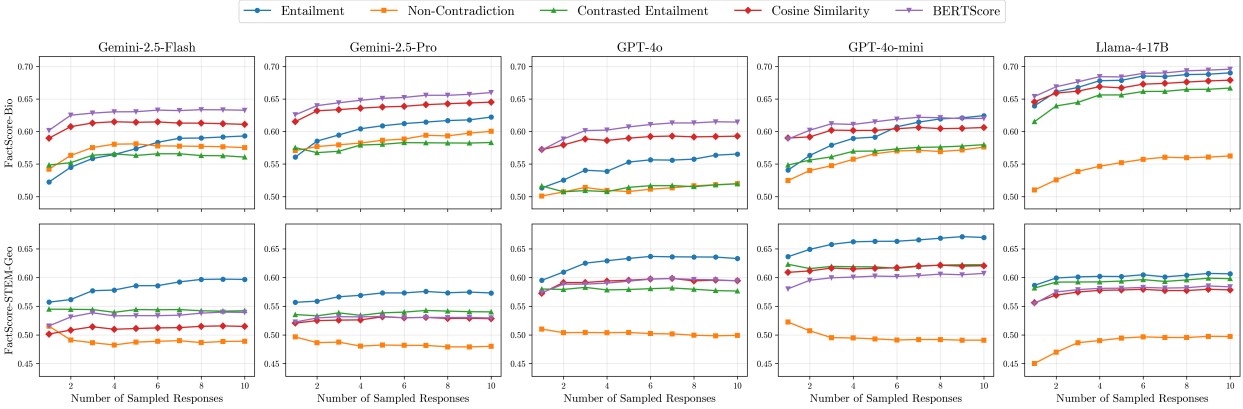

Figure 9: Matched-Sentence Hallucination Detection AUROC by Number of Sampled Responses

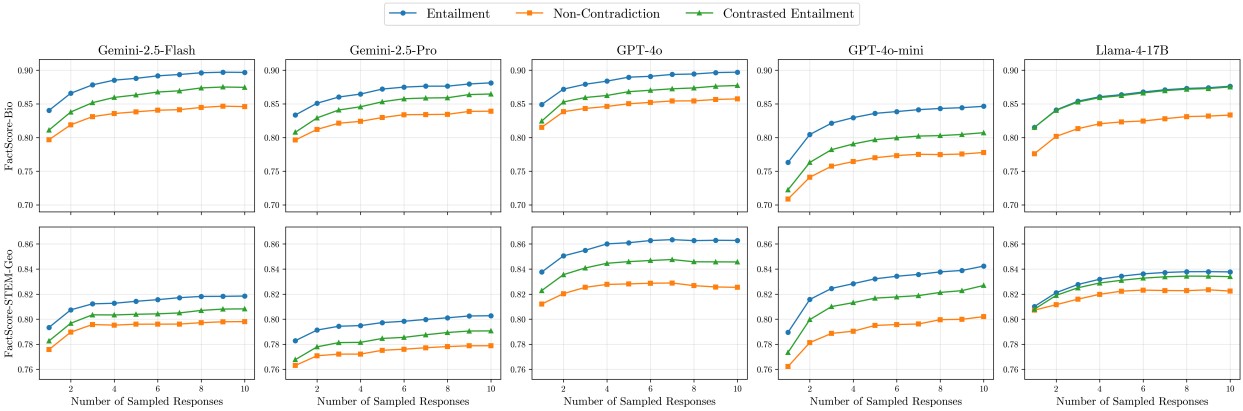

Figure 10: Claim-Response Hallucination Detection AUPRC by Number of Sampled Responses

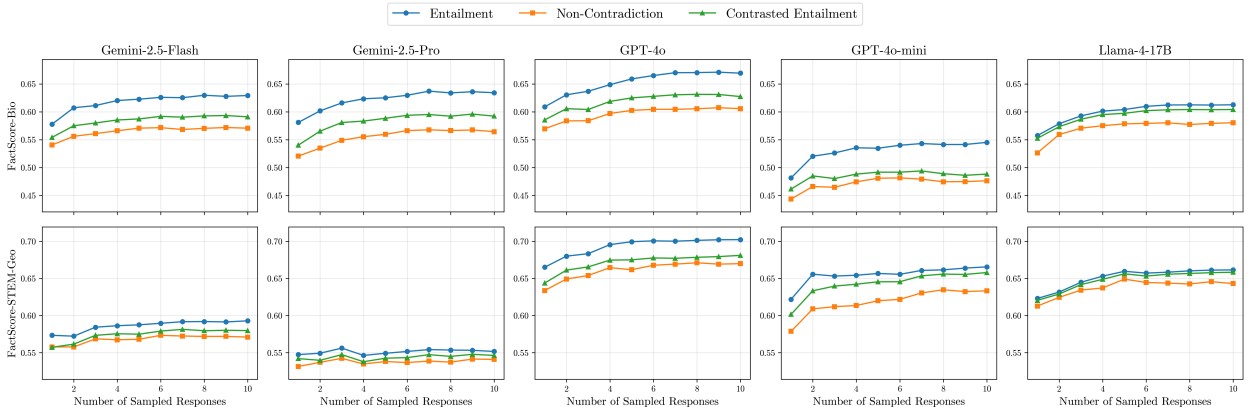

Figure 11: Sentence-Response Hallucination Detection AUPRC by Number of Sampled Responses

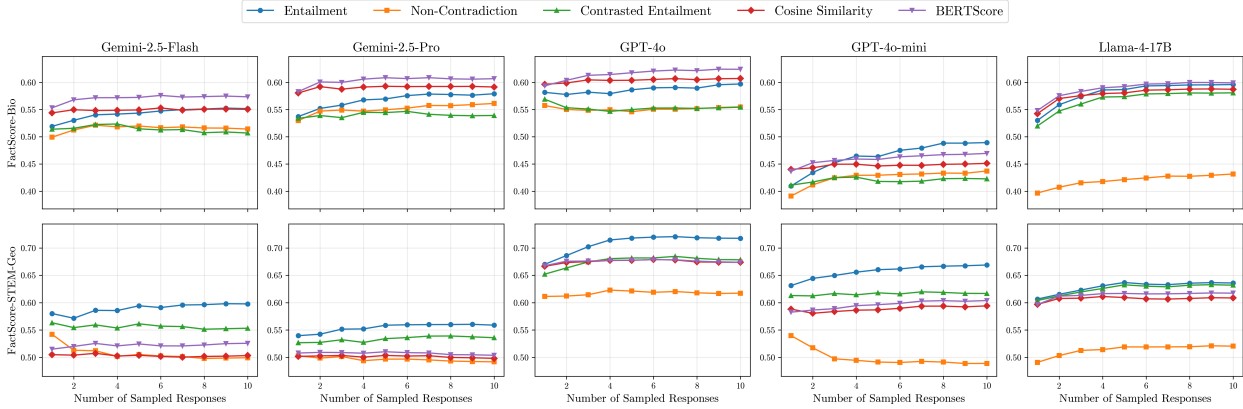

Figure 12: Matched-Sentence Hallucination Detection AUPRC by Number of Sampled Responses

# B Supplemental Tables and Figures

## B.1 Classification Results

Tables 11–14 provide the full claim-level and sentence-level classification results (AUROC and AUPRC) underlying the summary in Table 3 (Section 4.2). These tables report per-scorer, per-LLM results for both datasets, enabling detailed comparison across all consistency functions within each scorer family. Figure 13 displays AUPRC per-scorer, per-LLM for all scorers for both datasets.

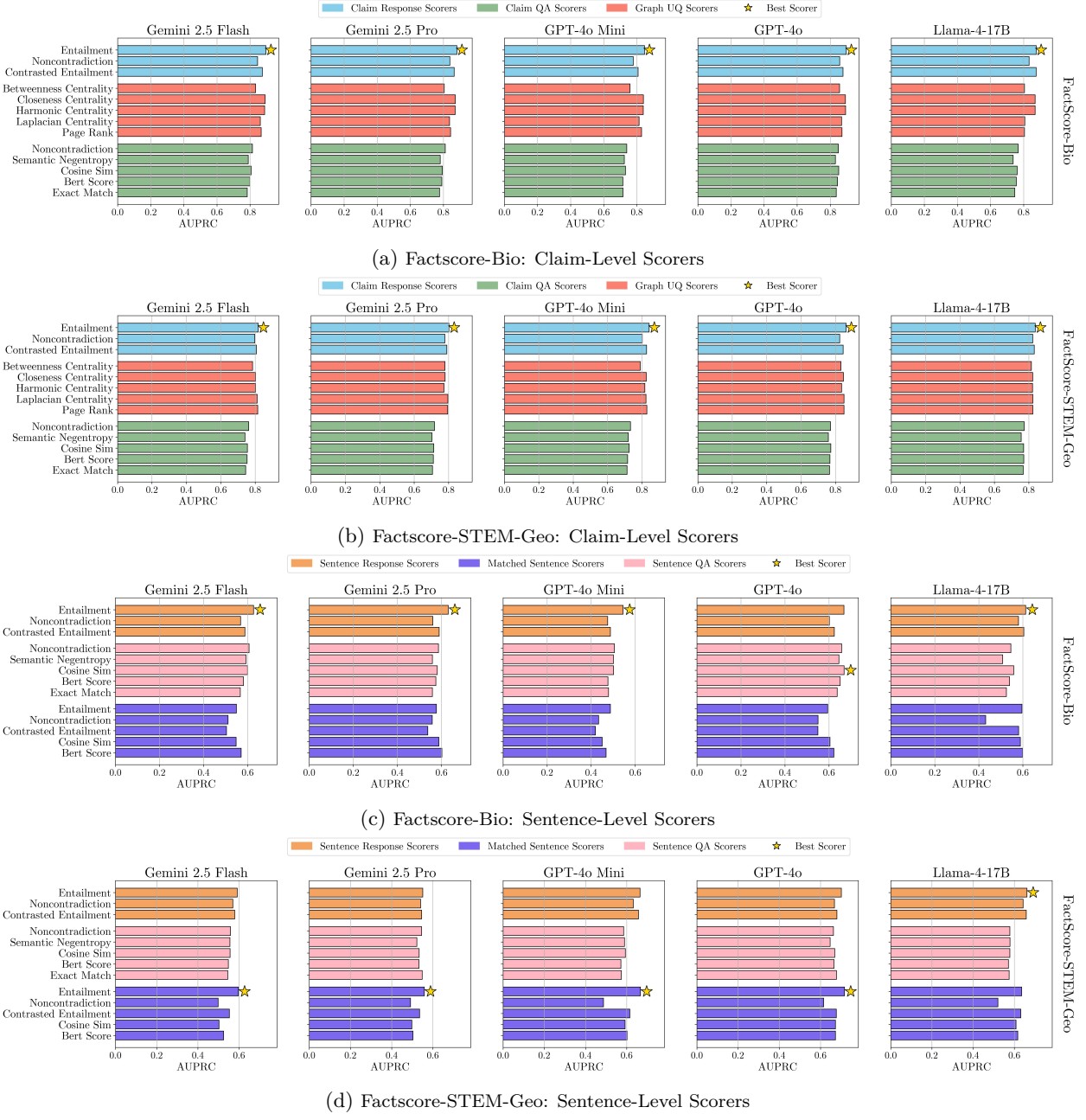

Figure 13: Claim-Level and Sentence-Level AUPRC by LLM and Dataset (Higher is Better)

| Scorer | Gemini-2.5 Flash | | Gemini-2.5 Pro | | GPT-4o Mini | | GPT-4o | | Llama-4-17B | |
|---|---|---|---|---|---|---|---|---|---|---|
| | ROC | PRC | ROC | PRC | ROC | PRC | ROC | PRC | ROC | PRC |
| **Claim-Response Scorers** | | | | | | | | | | |
| Entailment | 0.79 | **0.90** | 0.76 | **0.88** | 0.77 | **0.85** | 0.72 | **0.90** | **0.79** | **0.88** |
| Non-contradiction | 0.72 | 0.85 | 0.68 | 0.84 | 0.69 | 0.78 | 0.63 | 0.86 | 0.73 | 0.83 |
| Contrasted Entailment | 0.77 | 0.88 | 0.73 | 0.86 | 0.73 | 0.81 | 0.68 | 0.88 | 0.79 | 0.88 |
| **Claim-QA Scorers** | | | | | | | | | | |
| Exact Match | 0.60 | 0.78 | 0.58 | 0.78 | 0.60 | 0.72 | 0.60 | 0.84 | 0.60 | 0.75 |
| Semantic Entropy | 0.63 | 0.79 | 0.60 | 0.78 | 0.62 | 0.72 | 0.60 | 0.83 | 0.59 | 0.74 |
| Non-contradiction | 0.65 | 0.81 | 0.64 | 0.81 | 0.62 | 0.74 | 0.62 | 0.85 | 0.63 | 0.77 |
| BERTScore | 0.62 | 0.80 | 0.60 | 0.79 | 0.57 | 0.72 | 0.61 | 0.85 | 0.60 | 0.76 |
| Cosine Similarity | 0.64 | 0.81 | 0.61 | 0.79 | 0.61 | 0.73 | 0.62 | 0.85 | 0.61 | 0.76 |
| **Claim-Verbalized Confidence** | | | | | | | | | | |
| Verbalized Confidence | 0.65 | 0.79 | 0.59 | 0.77 | 0.66 | 0.75 | 0.61 | 0.84 | 0.69 | 0.79 |
| **Graph-Based Scorers** | | | | | | | | | | |
| Betweenness | 0.72 | 0.83 | 0.68 | 0.80 | 0.71 | 0.76 | 0.67 | 0.86 | 0.72 | 0.81 |
| Closeness | **0.80** | 0.89 | **0.76** | 0.87 | **0.78** | 0.84 | 0.73 | 0.89 | 0.79 | 0.87 |
| Harmonic | 0.80 | 0.89 | 0.76 | 0.87 | 0.78 | 0.84 | **0.73** | 0.89 | 0.79 | 0.87 |
| Page Rank | 0.77 | 0.87 | 0.72 | 0.84 | 0.77 | 0.83 | 0.69 | 0.87 | 0.73 | 0.81 |
| Laplacian | 0.75 | 0.86 | 0.71 | 0.84 | 0.75 | 0.81 | 0.69 | 0.87 | 0.73 | 0.81 |

Table 11: FactScore-Bio: Claim-Level Classification (AUROC↑, AUPRC↑) by LLM

| Scorer | Gemini-2.5 Flash | | Gemini-2.5 Pro | | GPT-4o Mini | | GPT-4o | | Llama-4-17B | |
|---|---|---|---|---|---|---|---|---|---|---|
| | ROC | PRC | ROC | PRC | ROC | PRC | ROC | PRC | ROC | PRC |
| **Sentence-Response Scorers** | | | | | | | | | | |
| Entailment | 0.68 | **0.63** | **0.69** | **0.63** | **0.70** | **0.55** | 0.65 | 0.67 | **0.72** | **0.61** |
| Non-contradiction | 0.62 | 0.57 | 0.62 | 0.56 | 0.66 | 0.47 | 0.60 | 0.60 | 0.67 | 0.58 |
| Contrasted Entailment | 0.65 | 0.59 | 0.66 | 0.59 | 0.67 | 0.49 | 0.61 | 0.63 | 0.70 | 0.60 |
| **Sentence-QA Scorers** | | | | | | | | | | |
| Exact Match | 0.63 | 0.57 | 0.64 | 0.56 | 0.64 | 0.48 | 0.63 | 0.64 | 0.63 | 0.53 |
| Semantic Negentropy | **0.68** | 0.59 | 0.65 | 0.56 | 0.69 | 0.50 | 0.65 | 0.65 | 0.63 | 0.51 |
| Non-contradiction | 0.68 | 0.61 | 0.67 | 0.59 | 0.69 | 0.51 | 0.65 | 0.66 | 0.65 | 0.55 |
| BERTScore | 0.64 | 0.58 | 0.65 | 0.58 | 0.63 | 0.48 | 0.64 | 0.65 | 0.65 | 0.54 |
| Cosine Similarity | 0.67 | 0.60 | 0.65 | 0.58 | 0.66 | 0.50 | **0.66** | **0.67** | 0.67 | 0.56 |
| **Sentence-Verbalized Confidence** | | | | | | | | | | |
| Verbalized Confidence | 0.61 | 0.52 | 0.56 | 0.49 | 0.59 | 0.43 | 0.58 | 0.59 | 0.67 | 0.54 |
| **Matched-Sentence Scorers** | | | | | | | | | | |
| Non-contradiction | 0.58 | 0.51 | 0.60 | 0.55 | 0.57 | 0.41 | 0.52 | 0.55 | 0.56 | 0.43 |
| Contrasted Entailment | 0.56 | 0.51 | 0.58 | 0.54 | 0.58 | 0.42 | 0.52 | 0.55 | 0.67 | 0.58 |
| Cosine Similarity | 0.61 | 0.55 | 0.65 | 0.59 | 0.61 | 0.45 | 0.59 | 0.61 | 0.68 | 0.59 |
| BERTScore | 0.63 | 0.57 | 0.66 | 0.61 | 0.62 | 0.47 | 0.62 | 0.62 | 0.70 | 0.60 |
| Entailment | 0.59 | 0.55 | 0.62 | 0.58 | 0.62 | 0.49 | 0.57 | 0.60 | 0.69 | 0.60 |

Table 12: FactScore-Bio: Sentence-Level Classification (AUROC↑, AUPRC↑) by LLM

| | Gemini-2.5 Flash | | Gemini-2.5 Pro | | GPT-4o Mini | | GPT-4o | | Llama-4-17B | |
|---|---|---|---|---|---|---|---|---|---|---|
| Scorer | ROC | PRC | ROC | PRC | ROC | PRC | ROC | PRC | ROC | PRC |
| **Claim-Response Scorers** | | | | | | | | | | |
| Entailment | 0.67 | **0.82** | 0.67 | **0.80** | **0.72** | **0.84** | **0.70** | **0.86** | **0.67** | **0.84** |
| Non-contradiction | 0.64 | 0.80 | 0.63 | 0.78 | 0.66 | 0.80 | 0.62 | 0.82 | 0.64 | 0.82 |
| Contrasted Entailment | 0.65 | 0.81 | 0.65 | 0.79 | 0.70 | 0.83 | 0.67 | 0.85 | 0.66 | 0.83 |
| **Claim-QA Scorers** | | | | | | | | | | |
| Exact Match | 0.57 | 0.74 | 0.54 | 0.71 | 0.55 | 0.71 | 0.55 | 0.77 | 0.57 | 0.77 |
| Semantic Negentropy | 0.57 | 0.74 | 0.54 | 0.70 | 0.57 | 0.72 | 0.54 | 0.76 | 0.55 | 0.76 |
| Non-contradiction | 0.59 | 0.76 | 0.56 | 0.72 | 0.58 | 0.73 | 0.55 | 0.77 | 0.58 | 0.77 |
| BERTScore | 0.57 | 0.75 | 0.54 | 0.71 | 0.54 | 0.72 | 0.55 | 0.77 | 0.57 | 0.77 |
| Cosine Similarity | 0.57 | 0.75 | 0.54 | 0.71 | 0.56 | 0.73 | 0.56 | 0.77 | 0.56 | 0.77 |
| **Claim-Verbalized Confidence** | | | | | | | | | | |
| Verbalized Confidence | 0.61 | 0.76 | 0.58 | 0.72 | 0.65 | 0.76 | 0.63 | 0.80 | 0.63 | 0.80 |
| **Graph-Based Scorers** | | | | | | | | | | |
| Betweenness | 0.65 | 0.78 | 0.66 | 0.78 | 0.68 | 0.79 | 0.68 | 0.83 | 0.65 | 0.82 |
| Closeness | 0.66 | 0.80 | 0.66 | 0.78 | 0.71 | 0.83 | 0.69 | 0.85 | 0.66 | 0.82 |
| Harmonic | 0.66 | 0.80 | 0.65 | 0.77 | 0.70 | 0.82 | 0.68 | 0.84 | 0.65 | 0.82 |
| Page Rank | **0.68** | 0.81 | **0.67** | 0.79 | 0.72 | 0.83 | 0.69 | 0.85 | 0.66 | 0.82 |
| Laplacian | 0.67 | 0.81 | 0.67 | 0.80 | 0.71 | 0.82 | 0.69 | 0.85 | 0.66 | 0.82 |

Table 13: FactScore-STEM-Geo: Claim-Level Classification (AUROC↑, AUPRC↑) by LLM

| | Gemini-2.5 Flash | | Gemini-2.5 Pro | | GPT-4o Mini | | GPT-4o | | Llama-4-17B | |
|---|---|---|---|---|---|---|---|---|---|---|
| Scorer | ROC | PRC | ROC | PRC | ROC | PRC | ROC | PRC | ROC | PRC |
| **Sentence-Response Scorers** | | | | | | | | | | |
| Entailment | 0.59 | 0.59 | 0.56 | 0.55 | 0.67 | 0.66 | 0.63 | 0.70 | **0.63** | **0.66** |
| Non-contradiction | 0.57 | 0.57 | 0.55 | 0.54 | 0.66 | 0.63 | 0.59 | 0.67 | 0.61 | 0.64 |
| Contrasted Entailment | 0.57 | 0.58 | 0.56 | 0.54 | **0.67** | 0.66 | 0.60 | 0.68 | 0.62 | 0.66 |
| **Sentence-QA Scorers** | | | | | | | | | | |
| Exact Match | 0.57 | 0.54 | 0.57 | 0.55 | 0.60 | 0.57 | 0.61 | 0.68 | 0.56 | 0.58 |
| Semantic Negentropy | 0.58 | 0.56 | 0.55 | 0.52 | 0.63 | 0.59 | 0.59 | 0.65 | 0.57 | 0.58 |
| Non-contradiction | 0.57 | 0.56 | 0.56 | 0.55 | 0.63 | 0.58 | 0.58 | 0.66 | 0.57 | 0.58 |
| BERTScore | 0.57 | 0.55 | 0.55 | 0.54 | 0.60 | 0.57 | 0.60 | 0.67 | 0.57 | 0.57 |
| Cosine Similarity | 0.57 | 0.56 | 0.54 | 0.53 | 0.62 | 0.59 | 0.60 | 0.67 | 0.57 | 0.58 |
| **Sentence-Verbalized Confidence** | | | | | | | | | | |
| Verbalized Confidence | 0.55 | 0.53 | 0.53 | 0.51 | 0.64 | 0.60 | **0.64** | 0.67 | **0.63** | 0.63 |
| **Matched-Sentence Scorers** | | | | | | | | | | |
| Non-contradiction | 0.49 | 0.50 | 0.48 | 0.49 | 0.49 | 0.49 | 0.50 | 0.60 | 0.50 | 0.52 |
| Contrasted Entailment | 0.54 | 0.55 | 0.54 | 0.54 | 0.62 | 0.61 | 0.58 | 0.68 | 0.60 | 0.63 |
| Cosine Similarity | 0.51 | 0.50 | 0.53 | 0.50 | 0.62 | 0.59 | 0.59 | 0.67 | 0.58 | 0.61 |
| BERTScore | 0.54 | 0.53 | 0.53 | 0.50 | 0.61 | 0.60 | 0.59 | 0.67 | 0.58 | 0.62 |
| Entailment | **0.60** | **0.60** | **0.57** | **0.56** | 0.67 | **0.67** | 0.63 | **0.72** | 0.61 | 0.62 |

Table 14: FactScore-STEM-Geo: Sentence-Level Classification (AUROC↑, AUPRC↑) by LLM

## B.2 Calibration Results

Tables 15–18 report the full calibration results (ECE and Brier Score) discussed in Section 4.2. These complement the classification results in Appendix B.1 by quantifying how well each scorer's confidence estimates align with observed factuality rates.

| Scorer | Gemini-2.5 Flash ECE | BS | Gemini-2.5 Pro ECE | BS | GPT-4o Mini ECE | BS | GPT-4o ECE | BS | Llama-4-17B ECE | BS |
|---|---|---|---|---|---|---|---|---|---|---|
| **Claim-Response Scorers** | | | | | | | | | | |
| Entailment | 0.12 | 0.18 | **0.12** | 0.19 | **0.10** | **0.19** | 0.15 | 0.19 | **0.10** | **0.19** |
| Non-contradiction | 0.19 | 0.22 | 0.20 | 0.22 | 0.27 | 0.28 | 0.16 | 0.18 | 0.61 | 0.62 |
| Contrasted Entailment | **0.10** | **0.17** | 0.12 | 0.19 | 0.18 | 0.23 | 0.12 | 0.17 | 0.30 | 0.28 |
| **Claim-QA Scorers** | | | | | | | | | | |
| Exact Match | 0.42 | 0.41 | 0.43 | 0.43 | 0.39 | 0.41 | 0.44 | 0.42 | 0.32 | 0.36 |
| Semantic Entropy | 0.16 | 0.22 | 0.18 | 0.22 | 0.21 | 0.27 | 0.17 | 0.20 | 0.23 | 0.29 |
| Non-contradiction | 0.18 | 0.22 | 0.21 | 0.23 | 0.27 | 0.30 | 0.17 | 0.19 | 0.29 | 0.31 |
| BERTScore | 0.22 | 0.24 | 0.22 | 0.24 | 0.32 | 0.33 | 0.17 | 0.19 | 0.31 | 0.32 |
| Cosine Similarity | 0.16 | 0.21 | 0.19 | 0.22 | 0.32 | 0.33 | 0.16 | 0.19 | 0.29 | 0.31 |
| **Claim-Verbalized Confidence** | | | | | | | | | | |
| Verbalized Confidence | 0.21 | 0.22 | 0.23 | 0.23 | 0.17 | 0.24 | **0.09** | 0.20 | 0.12 | 0.22 |
| **Graph-Based Scorers** | | | | | | | | | | |
| Betweenness | 0.71 | 0.70 | 0.72 | 0.72 | 0.63 | 0.63 | 0.78 | 0.77 | 0.63 | 0.62 |
| Closeness | 0.14 | 0.17 | 0.15 | **0.18** | 0.21 | 0.22 | 0.09 | **0.15** | 0.21 | 0.23 |
| Harmonic | 0.15 | 0.18 | 0.16 | 0.19 | 0.23 | 0.24 | 0.11 | 0.16 | 0.22 | 0.23 |
| Page Rank | 0.71 | 0.70 | 0.72 | 0.71 | 0.63 | 0.63 | 0.78 | 0.77 | 0.63 | 0.62 |
| Laplacian | 0.68 | 0.65 | 0.68 | 0.66 | 0.60 | 0.58 | 0.75 | 0.72 | 0.59 | 0.57 |

Table 15: FactScore-Bio: Claim-Level Calibration (ECE↓, BS↓) by LLM

| Scorer | Gemini-2.5 Flash ECE | BS | Gemini-2.5 Pro ECE | BS | GPT-4o Mini ECE | BS | GPT-4o ECE | BS | Llama-4-17B ECE | BS |
|---|---|---|---|---|---|---|---|---|---|---|
| **Sentence-Response Scorers** | | | | | | | | | | |
| Entailment | 0.16 | 0.26 | 0.16 | **0.25** | **0.13** | **0.23** | 0.17 | 0.27 | 0.15 | 0.24 |
| Non-contradiction | 0.47 | 0.46 | 0.50 | 0.49 | 0.55 | 0.53 | 0.41 | 0.41 | 0.42 | 0.42 |
| Contrasted Entailment | 0.34 | 0.36 | 0.37 | 0.38 | 0.43 | 0.41 | 0.30 | 0.34 | 0.15 | 0.24 |
| **Sentence-QA Scorers** | | | | | | | | | | |
| Exact Match | **0.10** | **0.25** | **0.11** | 0.25 | 0.16 | 0.25 | **0.09** | **0.25** | 0.20 | 0.28 |
| Semantic Negentropy | 0.37 | 0.36 | 0.43 | 0.41 | 0.43 | 0.39 | 0.31 | 0.33 | 0.41 | 0.41 |
| Non-contradiction | 0.43 | 0.41 | 0.48 | 0.46 | 0.53 | 0.49 | 0.38 | 0.38 | 0.51 | 0.49 |
| BERTScore | 0.50 | 0.49 | 0.51 | 0.51 | 0.62 | 0.60 | 0.43 | 0.43 | 0.55 | 0.54 |
| Cosine Similarity | 0.46 | 0.45 | 0.49 | 0.48 | 0.58 | 0.55 | 0.40 | 0.40 | 0.53 | 0.51 |
| **Sentence-Verbalized Confidence** | | | | | | | | | | |
| Verbalized Confidence | 0.45 | 0.44 | 0.48 | 0.48 | 0.46 | 0.43 | 0.28 | 0.32 | 0.34 | 0.33 |
| **Matched-Sentence Scorers** | | | | | | | | | | |
| Non-contradiction | 0.53 | 0.53 | 0.54 | 0.54 | 0.65 | 0.65 | 0.47 | 0.47 | 0.39 | 0.41 |
| Contrasted Entailment | 0.37 | 0.39 | 0.38 | 0.40 | 0.52 | 0.49 | 0.34 | 0.37 | 0.30 | 0.28 |
| Cosine Similarity | 0.40 | 0.40 | 0.42 | 0.41 | 0.52 | 0.49 | 0.34 | 0.36 | 0.29 | 0.31 |
| BERTScore | 0.46 | 0.45 | 0.47 | 0.46 | 0.58 | 0.56 | 0.39 | 0.40 | 0.31 | 0.32 |
| Entailment | 0.26 | 0.32 | 0.21 | 0.29 | 0.16 | 0.24 | 0.31 | 0.36 | **0.10** | **0.19** |

Table 16: FactScore-Bio: Sentence-Level Calibration (ECE↓, BS↓) by LLM

| Scorer | Gemini-2.5 Flash | | Gemini-2.5 Pro | | GPT-4o Mini | | GPT-4o | | Llama-4-17B | |
|---|---|---|---|---|---|---|---|---|---|---|
| | ECE | BS | ECE | BS | ECE | BS | ECE | BS | ECE | BS |
| **Claim-Response Scorers** | | | | | | | | | | |
| Entailment | 0.17 | 0.24 | **0.16** | **0.24** | **0.14** | 0.22 | 0.17 | 0.22 | 0.17 | 0.23 |
| Non-contradiction | 0.22 | 0.25 | 0.25 | 0.27 | 0.25 | 0.27 | 0.21 | 0.23 | 0.23 | 0.24 |
| Contrasted Entailment | **0.16** | 0.22 | 0.17 | 0.24 | 0.16 | 0.23 | 0.14 | 0.20 | 0.15 | 0.21 |
| **Claim-QA Scorers** | | | | | | | | | | |
| Exact Match | 0.39 | 0.40 | 0.38 | 0.40 | 0.46 | 0.47 | 0.47 | 0.46 | 0.28 | 0.30 |
| Semantic Negentropy | 0.19 | 0.25 | 0.22 | 0.27 | 0.21 | 0.27 | 0.21 | 0.24 | 0.24 | 0.21 |
| Non-contradiction | 0.23 | 0.26 | 0.27 | 0.29 | 0.26 | 0.28 | 0.22 | 0.24 | 0.24 | 0.25 |
| BERTScore | 0.24 | 0.26 | 0.27 | 0.29 | 0.28 | 0.30 | 0.22 | 0.24 | 0.24 | 0.25 |
| Cosine Similarity | 0.20 | 0.24 | 0.24 | 0.27 | 0.29 | 0.30 | 0.21 | 0.23 | 0.23 | 0.25 |
| **Claim-Verbalized Confidence** | | | | | | | | | | |
| Verbalized Confidence | 0.23 | 0.25 | 0.28 | 0.28 | 0.14 | **0.22** | **0.11** | 0.20 | **0.06** | **0.19** |
| **Graph-Based Scorers** | | | | | | | | | | |
| Betweenness | 0.70 | 0.69 | 0.68 | 0.67 | 0.67 | 0.66 | 0.73 | 0.72 | 0.72 | 0.72 |
| Closeness | 0.17 | **0.22** | 0.21 | 0.24 | 0.17 | 0.22 | 0.12 | **0.19** | 0.15 | 0.21 |
| Harmonic | 0.18 | 0.22 | 0.21 | 0.25 | 0.19 | 0.23 | 0.13 | 0.19 | 0.14 | 0.20 |
| Page Rank | 0.70 | 0.69 | 0.67 | 0.67 | 0.67 | 0.66 | 0.73 | 0.72 | 0.72 | 0.72 |
| Laplacian | 0.67 | 0.65 | 0.65 | 0.63 | 0.64 | 0.62 | 0.70 | 0.68 | 0.69 | 0.67 |

Table 17: FactScore-STEM-Geo: Claim-Level Calibration (ECE↓, BS↓) by LLM

| Scorer | Gemini-2.5 Flash | | Gemini-2.5 Pro | | GPT-4o Mini | | GPT-4o | | Llama-4-17B | |
|---|---|---|---|---|---|---|---|---|---|---|
| | ECE | BS | ECE | BS | ECE | BS | ECE | BS | ECE | BS |
| **Sentence-Response Scorers** | | | | | | | | | | |
| Entailment | 0.21 | 0.30 | 0.22 | 0.31 | 0.15 | **0.26** | 0.22 | 0.29 | 0.20 | 0.29 |
| Non-contradiction | 0.45 | 0.45 | 0.48 | 0.48 | 0.45 | 0.44 | 0.38 | 0.39 | 0.43 | 0.43 |
| Contrasted Entailment | 0.35 | 0.38 | 0.38 | 0.40 | 0.32 | 0.34 | 0.27 | 0.32 | 0.26 | 0.31 |
| **Sentence-QA Scorers** | | | | | | | | | | |
| Exact Match | **0.13** | **0.27** | **0.14** | **0.28** | **0.14** | 0.27 | **0.12** | **0.25** | 0.17 | 0.29 |
| Semantic Negentropy | 0.36 | 0.38 | 0.41 | 0.42 | 0.34 | 0.36 | 0.29 | 0.32 | 0.38 | 0.39 |
| Non-contradiction | 0.41 | 0.41 | 0.46 | 0.46 | 0.43 | 0.42 | 0.35 | 0.36 | 0.42 | 0.42 |
| BERTScore | 0.46 | 0.46 | 0.48 | 0.48 | 0.48 | 0.48 | 0.38 | 0.38 | 0.44 | 0.44 |
| Cosine Similarity | 0.43 | 0.43 | 0.46 | 0.46 | 0.46 | 0.45 | 0.36 | 0.36 | 0.42 | 0.42 |
| **Sentence-Verbalized Confidence** | | | | | | | | | | |
| Verbalized Confidence | 0.45 | 0.46 | 0.49 | 0.48 | 0.35 | 0.36 | 0.26 | 0.29 | 0.29 | 0.31 |
| **Matched-Sentence Scorers** | | | | | | | | | | |
| Non-contradiction | 0.50 | 0.49 | 0.51 | 0.51 | 0.51 | 0.51 | 0.41 | 0.41 | 0.23 | 0.24 |
| Contrasted Entailment | 0.35 | 0.38 | 0.36 | 0.39 | 0.35 | 0.37 | 0.27 | 0.32 | **0.15** | **0.21** |
| Cosine Similarity | 0.39 | 0.40 | 0.40 | 0.41 | 0.40 | 0.40 | 0.30 | 0.33 | 0.23 | 0.25 |
| BERTScore | 0.41 | 0.42 | 0.43 | 0.43 | 0.43 | 0.43 | 0.34 | 0.35 | 0.24 | 0.25 |
| Entailment | 0.25 | 0.32 | 0.25 | 0.32 | 0.23 | 0.29 | 0.29 | 0.33 | 0.17 | 0.23 |

Table 18: FactScore-STEM-Geo: Sentence-Level Calibration (ECE↓, BS↓) by LLM

## B.3 Correlation Results

Tables 19–21 provide the remaining response-level correlation results (Pearson and Spearman) for both datasets, supplementing Table 4 in the main text (Section 4.4). These tables include all scorer families, granularities, and short-form baselines, enabling full comparison of response-level scoring performance.

| Scorer | Gemini-2.5-Flash | Gemini-2.5-Pro | GPT-4o-Mini | GPT-4o | Llama-4-17B |
|---|---|---|---|---|---|
| **Claim-Level Scorers** | | | | | |
| CR Entailment | 0.58 | 0.52 | 0.66 | 0.44 | 0.64 |
| CR Non-Contradiction | 0.41 | 0.38 | 0.38 | 0.24 | 0.45 |
| CR Contrasted Entailment | 0.52 | 0.48 | 0.45 | 0.28 | 0.62 |
| QA Exact Match | 0.36 | 0.30 | 0.57 | 0.39 | 0.30 |
| QA Semantic Negentropy | 0.50 | 0.46 | 0.56 | 0.41 | 0.41 |
| QA Non-Contradiction | 0.50 | 0.52 | 0.41 | 0.44 | 0.57 |
| QA BERTScore | 0.43 | 0.41 | 0.41 | 0.40 | 0.35 |
| QA Cosine Similarity | 0.44 | 0.44 | 0.42 | 0.43 | 0.44 |
| Betweenness | -0.03 | -0.20 | -0.03 | -0.14 | -0.36 |
| Closeness | **0.61** | **0.54** | **0.69** | 0.45 | **0.65** |
| Harmonic | 0.59 | 0.53 | 0.67 | **0.46** | 0.63 |
| Page Rank | 0.42 | 0.34 | 0.56 | 0.25 | 0.23 |
| Laplacian | 0.39 | 0.29 | 0.51 | 0.23 | 0.20 |
| **Sentence-Level Scorers** | | | | | |
| SR Entailment | 0.34 | 0.34 | 0.47 | 0.22 | 0.36 |
| SR Non-Contradiction | 0.35 | 0.31 | 0.39 | 0.30 | 0.37 |
| SR Contrasted Entailment | 0.33 | 0.32 | 0.37 | 0.18 | 0.33 |
| Matched Entailment | 0.34 | 0.31 | 0.51 | 0.16 | 0.28 |
| Matched Non-Contradiction | 0.16 | 0.22 | 0.14 | -0.03 | 0.32 |
| Matched Contrasted Entailment | 0.20 | 0.25 | 0.19 | -0.01 | 0.22 |
| Matched Cosine Similarity | 0.31 | 0.30 | 0.44 | 0.19 | 0.40 |
| Matched BERTScore | 0.37 | 0.34 | 0.49 | 0.24 | 0.43 |
| QA Exact Match | 0.31 | 0.26 | 0.55 | 0.37 | 0.38 |
| QA Semantic Negentropy | 0.44 | 0.32 | 0.55 | 0.39 | 0.37 |
| QA Non-Contradiction | 0.42 | 0.36 | 0.49 | 0.30 | 0.40 |
| QA BERTScore | 0.38 | 0.29 | 0.49 | 0.37 | 0.45 |
| QA Cosine Similarity | 0.40 | 0.30 | 0.55 | 0.40 | 0.51 |
| **Short-Form Scorers** | | | | | |
| Semantic Negentropy | 0.39 | 0.28 | 0.42 | 0.26 | 0.35 |
| Non-Contradiction | 0.42 | 0.32 | 0.32 | 0.28 | 0.50 |
| BERTScore | 0.43 | 0.44 | 0.66 | 0.42 | 0.52 |
| Cosine Similarity | 0.39 | 0.37 | 0.49 | 0.22 | 0.39 |
| Normalized Sequence Probability | 0.02 | 0.01 | 0.60 | 0.49 | – |

Table 19: Spearman correlation ↑ between response-level confidence and factuality on FactScore-Bio by LLM

| Scorer | Gemini-2.5-Flash | Gemini-2.5-Pro | GPT-4o-Mini | GPT-4o | Llama-4-17B |
|---|---|---|---|---|---|
| **Claim-Level Scorers** | | | | | |
| CR Entailment | 0.34 | 0.20 | 0.50 | 0.23 | 0.09 |
| CR Non-Contradiction | 0.48 | **0.39** | 0.58 | 0.34 | 0.38 |
| CR Contrasted Entailment | 0.39 | 0.26 | 0.46 | 0.15 | 0.18 |
| QA Exact Match | 0.40 | 0.18 | 0.33 | 0.09 | 0.39 |
| QA Semantic Negentropy | 0.45 | 0.26 | 0.40 | 0.15 | 0.34 |
| QA Non-Contradiction | **0.54** | 0.38 | 0.61 | **0.37** | **0.48** |
| QA BERTScore | 0.39 | 0.17 | 0.03 | -0.03 | 0.30 |
| QA Cosine Similarity | 0.38 | 0.15 | 0.15 | 0.13 | 0.29 |
| Betweenness | -0.02 | 0.03 | 0.07 | -0.02 | -0.13 |
| Closeness | 0.39 | 0.23 | 0.52 | 0.26 | 0.14 |
| Harmonic | 0.36 | 0.22 | 0.44 | 0.16 | 0.11 |
| Page Rank | 0.34 | 0.25 | 0.47 | 0.22 | 0.08 |
| Laplacian | 0.32 | 0.23 | 0.45 | 0.19 | 0.04 |
| **Sentence-Level Scorers** | | | | | |
| SR Entailment | 0.02 | -0.06 | 0.19 | -0.02 | 0.16 |
| SR Non-Contradiction | 0.35 | 0.29 | 0.37 | 0.24 | 0.37 |
| SR Contrasted Entailment | 0.19 | 0.09 | 0.20 | -0.03 | 0.11 |
| Matched Entailment | 0.02 | -0.12 | 0.19 | 0.07 | 0.13 |
| Matched Non-Contradiction | 0.22 | -0.02 | -0.03 | -0.03 | 0.16 |
| Matched Contrasted Entailment | 0.05 | -0.02 | -0.06 | -0.12 | 0.09 |
| Matched Cosine Similarity | 0.02 | 0.03 | 0.25 | 0.04 | 0.11 |
| Matched BERTScore | 0.12 | -0.06 | 0.20 | 0.07 | 0.36 |
| QA Exact Match | 0.17 | 0.19 | 0.41 | 0.23 | 0.31 |
| QA Semantic Negentropy | 0.27 | 0.11 | 0.34 | 0.22 | 0.35 |
| QA Non-Contradiction | 0.25 | 0.18 | 0.41 | 0.30 | 0.38 |
| QA BERTScore | 0.20 | 0.17 | 0.40 | 0.20 | 0.36 |
| QA Cosine Similarity | 0.22 | 0.11 | 0.39 | 0.20 | 0.40 |
| **Short-Form Scorers** | | | | | |
| Semantic Negentropy | 0.03 | -0.02 | 0.31 | -0.05 | -0.02 |
| Non-Contradiction | 0.36 | 0.24 | **0.62** | 0.31 | 0.29 |
| BERTScore | 0.17 | -0.08 | 0.21 | 0.04 | -0.22 |
| Cosine Similarity | 0.14 | -0.02 | 0.14 | -0.02 | 0.02 |
| Normalized Sequence Probability | 0.11 | 0.00 | 0.50 | 0.27 | – |

Table 20: Pearson correlation coefficients (higher is better) between response-level confidence scores and factuality on FactScore-STEM-Geo by LLM.

| Scorer | Gemini-2.5-Flash | Gemini-2.5-Pro | GPT-4o-Mini | GPT-4o | Llama-4-17B |
|---|---|---|---|---|---|
| **Claim-Level Scorers** | | | | | |
| CR Entailment | 0.25 | 0.15 | 0.42 | 0.15 | 0.06 |
| CR Non-Contradiction | 0.44 | **0.38** | 0.50 | 0.32 | 0.29 |
| CR Contrasted Entailment | 0.30 | 0.20 | 0.36 | 0.09 | 0.10 |
| QA Exact Match | 0.40 | 0.16 | 0.33 | 0.07 | 0.36 |
| QA Semantic Negentropy | 0.37 | 0.18 | 0.37 | 0.12 | 0.27 |
| QA Non-Contradiction | **0.48** | 0.33 | **0.51** | **0.34** | **0.46** |
| QA BERTScore | 0.37 | 0.13 | 0.08 | 0.00 | 0.27 |
| QA Cosine Similarity | 0.30 | 0.11 | 0.18 | 0.16 | 0.23 |
| Betweenness | 0.01 | 0.06 | 0.11 | -0.02 | -0.08 |
| Closeness | 0.25 | 0.15 | 0.40 | 0.16 | 0.10 |
| Harmonic | 0.25 | 0.16 | 0.35 | 0.09 | 0.11 |
| Page Rank | 0.31 | 0.23 | 0.43 | 0.15 | 0.04 |
| Laplacian | 0.28 | 0.21 | 0.41 | 0.14 | 0.00 |
| **Sentence-Level Scorers** | | | | | |
| SR Entailment | -0.04 | -0.12 | 0.15 | -0.05 | 0.13 |
| SR Non-Contradiction | 0.27 | 0.20 | 0.21 | 0.15 | 0.31 |
| SR Contrasted Entailment | 0.09 | -0.03 | 0.08 | -0.08 | 0.08 |
| Matched Entailment | -0.02 | -0.14 | 0.17 | 0.05 | 0.11 |
| Matched Non-Contradiction | 0.14 | 0.05 | 0.16 | 0.16 | 0.31 |
| Matched Contrasted Entailment | -0.01 | -0.10 | -0.11 | -0.15 | 0.08 |
| Matched Cosine Similarity | -0.06 | -0.05 | 0.19 | 0.02 | 0.10 |
| Matched BERTScore | 0.05 | -0.10 | 0.16 | 0.05 | 0.08 |
| QA Exact Match | 0.16 | 0.21 | 0.38 | 0.25 | 0.29 |
| QA Semantic Negentropy | 0.24 | 0.05 | 0.30 | 0.18 | 0.27 |
| QA Non-Contradiction | 0.20 | 0.10 | 0.36 | 0.24 | 0.31 |
| QA BERTScore | 0.17 | 0.15 | 0.37 | 0.19 | 0.34 |
| QA Cosine Similarity | 0.20 | 0.08 | 0.35 | 0.16 | 0.36 |
| **Short-Form Scorers** | | | | | |
| Semantic Negentropy | -0.09 | -0.08 | 0.21 | -0.12 | -0.06 |
| Non-Contradiction | 0.28 | 0.05 | 0.45 | 0.18 | 0.21 |
| BERTScore | 0.05 | -0.14 | 0.15 | 0.03 | -0.26 |
| Cosine Similarity | -0.03 | -0.09 | 0.00 | -0.09 | -0.02 |
| Normalized Sequence Probability | 0.11 | 0.03 | 0.45 | 0.24 | – |

Table 21: Spearman correlation ↑ between response-level confidence and factuality on FactScore-STEM-Geo by LLM

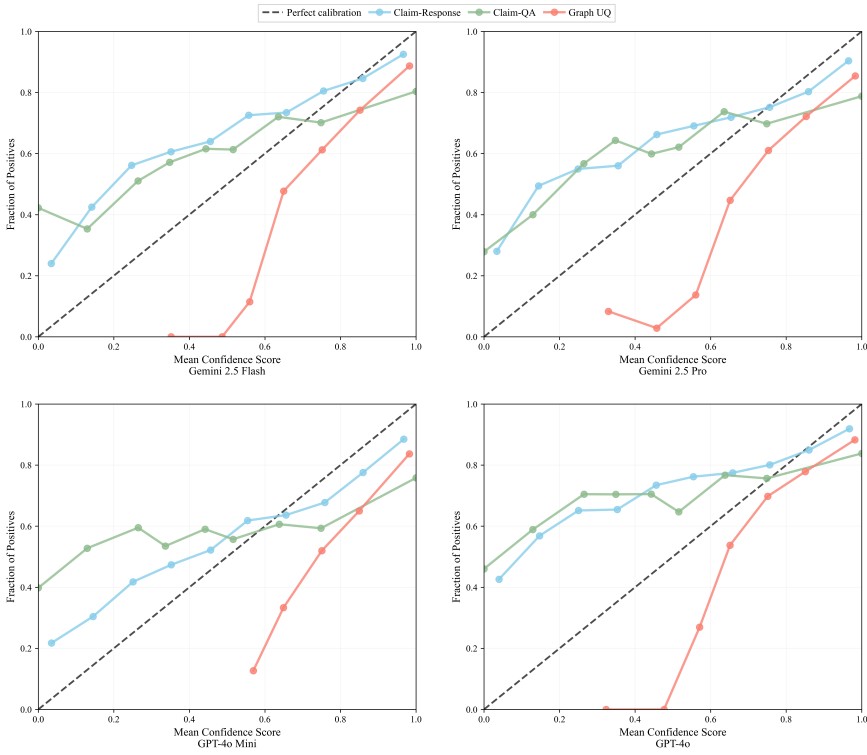

Figure 14: Claim-Level Calibration Plots (Top Scorer per Family) for FactScore-Bio by LLM

### B.4 Data and Runtime Information

Tables 22 and 23 report descriptive statistics of the generated responses (sentence counts, claim counts, response length, and claim diversity ratios) and approximate computational costs (semantic comparisons per prompt and runtime per dataset) referenced throughout Sections 4.1 and 5.

|  |  | Avg. Sentences per Response | Avg. Claims per Response | Avg. Words per Response | Claim Diversity Ratio |
|---|---|---|---|---|---|
| FactScore Bio | GPT-4o | $5.37 \pm 0.07$ | $24.30 \pm 0.41$ | $95.58 \pm 0.62$ | $2.72 \pm 0.11$ |
|  | GPT-4o-Mini | $5.19 \pm 0.05$ | $23.31 \pm 0.36$ | $95.95 \pm 0.27$ | $3.21 \pm 0.10$ |
|  | Gem.-2.5 Flash | $4.43 \pm 0.06$ | $21.82 \pm 0.44$ | $80.06 \pm 0.68$ | $3.03 \pm 0.13$ |
|  | Gem.-2.5 Pro | $4.78 \pm 0.06$ | $23.21 \pm 0.37$ | $88.08 \pm 0.38$ | $2.62 \pm 0.12$ |
|  | Llama-4-17B | $5.40 \pm 0.08$ | $21.65 \pm 0.43$ | $88.60 \pm 0.80$ | $2.79 \pm 0.10$ |
| FactScore STEM-Geo | GPT-4o | $6.24 \pm 0.09$ | $28.57 \pm 0.53$ | $142.75 \pm 1.48$ | $2.82 \pm 0.05$ |
|  | GPT-4o-Mini | $5.49 \pm 0.07$ | $27.52 \pm 0.47$ | $140.54 \pm 1.42$ | $3.05 \pm 0.07$ |
|  | Gem.-2.5 Flash | $4.38 \pm 0.07$ | $29.11 \pm 0.60$ | $131.99 \pm 1.97$ | $2.62 \pm 0.07$ |
|  | Gem.-2.5 Pro | $4.98 \pm 0.07$ | $32.09 \pm 0.54$ | $155.80 \pm 1.92$ | $2.24 \pm 0.04$ |
|  | Llama-4-17B | $5.57 \pm 0.07$ | $25.37 \pm 0.62$ | $135.43 \pm 1.43$ | $2.45 \pm 0.05$ |

Table 22: Descriptive Statistics of Long-Text Responses: Granular Units and Response Length. Claim Diversity Ratio is $N_{m\text{-union}}/N_{\text{claim}}$, the ratio of unique claims across sampled responses to claims in the original response; higher values indicate greater response diversity.

| Scorer Family | Approx. Comparisons/Prompt | Approx. Runtime/Dataset |
|---|---|---|
| Claim-Response | 250 | 2.5 hours |
| Graph-Based | 750 | 7.5 hours |
| Claim-QA | 250 | 2.5 hours |
| Sentence-response | 50 | 0.5 hours |
| Matched-Sentence | 250 | 2.5 hours |
| Sentence-QA | 50 | 0.5 hours |

Table 23: Approximate semantic comparisons per prompt and runtime per LLM-dataset. Runtimes measured using `microsoft/deberta-large-mnli` on a single NVIDIA T4 GPU. LLM generation time is excluded, as it varies with deployment-specific rate limits and hardware configurations.

## C FactScore-STEM-Geo Construction

We use the Wikipedia API to construct long-form QA datasets spanning four topics: Chemical Elements, Scientific Laws, Nerves in the human body, and Mountains with the goal of spanning four domains: chemistry, math/physics, biology/anatomy, and geography/geology, respectively. For each topic, we select the 100 Wikipedia entries with the longest texts to ensure sufficient content for grading LLM responses. The code to construct these datasets is given below:

```python
import wikipediaapi
def get_wiki_texts_from_entities(
    entities: List[str]
) -> List[str]:
    wiki_wiki = wikipediaapi.Wikipedia(
        user_agent=USER_AGENT,
        language='en'
    )
    texts = {}
    for entity in entities:
        page = wiki_wiki.page(entity).text
        if page:
            texts[entity] = page.text

    if len(texts) > 100:
        sorted_entities = sorted(
```

```
        texts.keys(),
        key=lambda x: len(texts[x]),
        reverse=True
    )[:100]
    top_texts = {}
    for entity in sorted_entities:
        top_texts[entity] = texts[entity]

return top_texts
```

A comprehensive list of the entities can be found in the supplemental source code.

# D   Prompt Templates

Prompt templates were adapted and refined from previously published protocols and are provided to support reproducibility. User prompts for FactScore grading and claim merging are adapted from Jiang et al. (2024), the user prompt for claim decomposition is adapted from Zhang et al. (2025), the user prompt for unit-QA question creation is adapted from Farquhar et al. (2024), and the user prompt for verbalized confidence is adapted from Tian et al. (2023); Jiang et al. (2024).

```
You are a precise and objective fact-checking assistant specialized in evaluating factual claims
    against provided context. Your task is to determine whether claims are supported by the given
    context, following the FactScore evaluation protocol.

Guidelines for your evaluations:
1. Analyze each claim strictly based on the provided context, not your prior knowledge
2. Respond with "Yes" only if the claim is directly supported by information in the context
3. Respond with "No" if:
   - The claim contradicts the context
   - The claim contains information not present in the context
   - The claim makes assertions that go beyond what the context states

Important principles:
- Be conservative in your judgments - only mark claims as supported when there is clear evidence
- Ignore stylistic differences or paraphrasing if the factual content matches
- Do not make assumptions or inferences beyond what is explicitly stated in the context
- Maintain consistency in your evaluation criteria across all claim-context pairs

Your responses should be limited to "Yes" or "No" without additional explanation, as these will be
    processed automatically in the FactScore evaluation framework.
```

Listing 1: System prompt for FactScore grading

```
Context: {answer}
Claim: {claim}
Is the claim supported by the context above?
Answer only Yes or No:
```

Listing 2: User Prompt for FactScore Grading

```
You are an expert linguistic analyst specializing in distinguishing between objective and
    subjective statements.

Objective statements present verifiable facts, information, or observations that can be proven
    true or false through evidence. They are independent of personal interpretations or biases.
    Examples include statistical data, historical events, scientific measurements, or established
    facts.
```

```
Subjective statements express judgments, evaluations, or perspectives that may vary between
    individuals. They cannot be definitively proven true or false as they depend on viewpoint,
    taste, or interpretation. Examples include value judgments, aesthetic assessments, or
    statements containing evaluative language.

When analyzing a statement, consider:
- Does it contain verifiable facts or measurable data?
- Does it include evaluative terms like "good," "beautiful," "better," "worst," "important," or
    "significant"?
- Could different people reasonably disagree about the statement?
- Is the statement presenting information that exists independent of human judgment?

Respond only with "objective" or "subjective" based on your analysis.
```

Listing 3: System Prompt for Objectivity Classification

```
Input: {claim}
Is the input subjective or objective?
Answer only subjective or objective:
```

Listing 4: User prompt for Objectivity Classification

```
Please breakdown the following passage into independent fact pieces.

Step 1: For each sentence, you should break it into several fact pieces. Each fact piece
should only contain one single independent fact. Normally the format of a fact piece is
"subject + verb + object". If the sentence does not contain a verb, you can use "be" as the
verb.

Step 2: Do this for all the sentences. Output each piece of fact in one single line starting
with ###. Do not include other formatting.

Step 3: Each atomic fact should be self-contained. Do not use pronouns as the subject of a
piece of fact, such as he, she, it, this that, use the original subject whenever possible.

Step 4: If the sentence does not contain any independent fact, you should output "### NONE".

Here are some examples:

Example 1:
Michael Collins (born October 31, 1930) is a retired American astronaut and test pilot who was
the Command Module Pilot for the Apollo 11 mission in 1969.
### Michael Collins was born on October 31, 1930.
### Michael Collins is retired.
### Michael Collins is an American.
### Michael Collins was an astronaut.
### Michael Collins was a test pilot.
### Michael Collins was the Command Module Pilot.
### Michael Collins was the Command Module Pilot for the Apollo 11 mission.
### Michael Collins was the Command Module Pilot for the Apollo 11 mission in 1969.

Example 2:
League of Legends (often abbreviated as LoL) is a multiplayer online battle arena (MOBA) video
game developed and published by Riot Games.
### League of Legends is a video game.
### League of Legends is often abbreviated as LoL.
### League of Legends is a multiplayer online battle arena.
### League of Legends is a MOBA video game.
```

```
    ### League of Legends is developed by Riot Games.
    ### League of Legends is published by Riot Games.

    Example 3:
    Emory University has a strong athletics program, competing in the National Collegiate Athletic
      Association (NCAA) Division I Atlantic Coast Conference (ACC). The university's mascot is the
      Eagle.
    ### Emory University has a strong athletics program.
    ### Emory University competes in the National Collegiate Athletic Association Division I.
    ### Emory University competes in the Atlantic Coast Conference.
    ### Emory University is part of the ACC.
    ### Emory University's mascot is the Eagle.

    Example 4:
    Hi
    ### NONE

    Now it's your turn. Here is the passage:

    {response}

    You should only return the final answer. Now your answer is:
```

Listing 5: User prompt for claim decomposition

```
We are writing an answer to this question: {original_question}

Describe how likely it is that the specific claim below is correct as one of the following
    expressions:

No chance (0%)
Little chance (20%)
Less than even (40%)
Fairly possible (60%)
Very good chance (80%)
Almost certain (100%)

Give ONLY your confidence phrase, no other words or explanation. Your answer must contain ONLY one
    of the confidence phrases above.

Here is the claim: {claim}

Now your answer is:
```

Listing 6: User prompt for Verbalized Confidence

```
Following this text: {response}
You see the sentence: {factoid}
Generate a list of {num_questions} questions, that might have generated the sentence in the
    context of the preceding original text. Please do not use specific facts that appear in the
    follow-up sentence when formulating the questions. Avoid yes-no questions. The questions
    should have a concise, well-defined answer e.g. only a name, place, or thing. Output each
    question in one single line starting with ###. Do not include other formatting.

Example:
### Here is the first question? ### Here is the second question?
```

```
Now your response is:
```

Listing 7: User prompt for Unit-QA question creation

```
Given two lists titled "Original Claim List" and "New Claim List", your task is to integrate
    information from the "New Claim List" into the "Original Claim List". Please follow these
    detailed steps to ensure accuracy and clarity in the process:
Task 1. **Verification Process:** Your goal is to go through each statement in the "New Claim
    List" one by one, and determine if it is fully entailed or mentioned by any statement in the
    "Original Claim List."
Task 2. **Compilation of Non-Entailed Claims:** Generate a list of statements from the "New Claim
    List" that are not already covered or implied by the "Original Claim List." For each new or
    unique claim that does not have an equivalent in the original list, format your output by
    starting each line with a dash ('-').
**Original Claim List:**
{master_claim_str}
**New Claim List:**
{sampled_claim_str}
Begin with the Verification Process to assess each claim's relevance and uniqueness, followed by
    the Compilation of Non-Entailed Claims to clearly list any new insights that the "New Claim
    List" provides.
```

Listing 8: User prompt for Claim Merging

