# OpenReview forum: "Fine-Grained Uncertainty Quantification for Long-Form Language Model Outputs: A Comparative Study"
_TMLR — Accepted by TMLR_

### Review · Reviewer_eJQG · 2026-03-16

**Summary Of Contributions:**

This paper introduces a for long-form black-box uncertainty quantification (UQ). This kind of UQ addresses the limitations of detecting hallucinations in multi-sentence or multi-claim responses by decomposing content into granular units.
All observed methods follow the three stages: response decomposition into sentences or atomic claims, confidence scoring based on semantic consistency, and score aggregation.
The presented taxonomy is divided four families that are standardizing methodologies.
The authors experiment across four LLMs (gemini and gpt families) and two datasets, including one of their own creation.


### Strengths

1. The unified taxonomy and the design standardization for UQ are novel and much needed in the field.
2. Figures 1-4 provide great visual aid for each family description.
3. The paper demonstrates a great variety of experimental settings and a new dataset for long-form UQ.


### Weaknesses

1. The utilized LLMs are relatively close in terms of performance and biases. Making models vary more would have made results more convincing.
2. While the quantity of experiments looks good, the quality is concerning, given that some methods were only "moderately calibrated."

**Audience:**

Yes

**Audience Explanation:**

This work presents a structured taxonomy of UQ. It will be of interest for AI researchers in related areas. For instance, researchers in hallucination detection.

**Broader Impact Concerns:**

No ethical concerns

**Claims And Evidence:**

Yes

**Claims Explanation:**

All findings have direct evidence in the experimental section.

**Requested Changes:**

### Critical

The writing of the main section left me under impression that each of four families is represented by a single model. However, judging by the Section 2, it is not true. Could authors emphasize in section 1 and/or section 3 that other methods are in the family?

The models are relatively close to each other within a family of models. I understand that it takes time an money to run such experiments. So i request to include **one** more non-thinking LLM that is not from Gemini or GPT family. You may look to Qwen or LLaMa.

Speaking of thinking, how do your findings hold when model is a thinking one? For example, o3-mini or Qwen-3-thinking. Which UQ families can operate with responses of the thinking model? This result is interesting, because thinking models tend to be very self-confident, so you have less variety in responses.

Can authors explain the "prohibitive" costs argument for matched claims? In your code you seem to be sensibly caching all the computation expensive steps like LLM responses. I can see that underlying anonlib you are using don't use batching of claims. Is that the only reason?




### Minor

It would improve the quality of work if references to the appendixes are introduced into the text. For example, indication that there is an ablation study or how FactScore-STEM-Geo is constructed.

It surprises me that neither introduction nor abstract mention the FactScore-STEM-Geo that you made. I recommend to stress out that fact. The new dataset is a strong contribution.

---

> ### Author Response · Authors · 2026-05-01
> **Revisions made with all suggestions added**
>
> We thank the reviewer for their constructive feedback and thoughtful suggestions. Below we address each point in detail.
>
> ## Critical Items
>
> ### 1. Emphasize that each family contains multiple methods
>
> We agree this was not sufficiently clear. We have made two additions:
>
> - In Section 1, we now state: "Within this framework, we define several families of fine-grained scorers, each generalizing one or more existing methods."
> - In Section 3.3, we now state: "Each family encompasses one or more existing methods as special cases and admits extensions through alternative semantic consistency functions and granularities."
>
> We have also added footnotes in Sections 3.1, 3.3.3, and 3.3.4 clarifying that the LLMs used for auxiliary tasks (decomposition, question generation, claim merging) need not be the same as the LLM used for response generation, and we have made the distinction between target LLMs and auxiliary LLMs more explicit in Section 4.1.
>
> ### 2. Include one more non-thinking LLM outside Gemini/GPT
>
> We have added Llama-4-Maverick-17B (Meta), an open-weight, non-thinking model, as a fifth LLM. We ran all scorers across both datasets and both granularities. The results are integrated into all applicable tables and figures. Results are highly consistent with the existing four models, confirming all three headline findings.
>
> ### 3. How do findings hold for thinking models?
>
> Our experiments already include two thinking models: Gemini-2.5-Pro and Gemini-2.5-Flash, both of which use chain-of-thought reasoning by default in the Gemini 2.5 series. We address the reviewer's specific concerns:
>
> **Do the findings hold?** Yes. The Gemini 2.5 models produce results that are consistent with the non-thinking models (GPT-4o, GPT-4o-mini, Llama-4) across all scorer families, evaluation metrics, and datasets. The relative ranking of scorer families is preserved, and claim-response entailment remains a top performer.
>
> **Is response diversity reduced?** To directly assess this, we now report the claim diversity ratio (N_m-union / N_claim) per model in Table 15. This ratio measures the number of unique claims across sampled responses relative to claims in the original response. Gemini-2.5-Pro exhibits the lowest diversity ratio across both datasets (2.62 on FactScore-Bio, 2.24 on FactScore-STEM-Geo), consistent with the reviewer's intuition that thinking models may produce more self-consistent outputs. However, this reduced diversity does not meaningfully degrade scorer performance.
>
> **Which UQ families can operate with thinking models?** All four families operate without modification, as they are black-box methods that require only sampled text responses. The thinking trace is internal to the model and does not affect the scoring pipeline.
>
> We discuss these observations in Section 7 (Limitations) and note that a more targeted study varying thinking budgets and evaluating dedicated reasoning models (e.g., o3, QwQ) remains an interesting direction for future work.
>
> ### 4. Explain the "prohibitive" costs for matched claims
>
> We have expanded footnote 9 to provide a concrete cost analysis: with N_claim ≈ 25 claims per response and m = 10 sampled responses, matched-claim scoring requires approximately m · N_claim² = 6,250 NLI comparisons per prompt, roughly 25x the cost of matched-sentence scoring.
>
> Regarding the reviewer's question about caching and batching: caching avoids redundant LLM generation calls but does not reduce the number of NLI comparisons, which is the primary bottleneck. The quadratic scaling in the number of claims per response (N_claim²) means that even with efficient batching of NLI inference, the total computation remains approximately 25x that of matched-sentence scoring. Across 900 prompts and 5 LLMs, this renders matched-claim experiments impractical with our available hardware (single NVIDIA T4 GPU for NLI inference). We also note that graph-based scorers already decompose sampled responses and achieve comparable performance to claim-response methods, so matched-claim scoring is unlikely to provide additional insight beyond what our existing experiments cover.
>
> ## Minor Items
>
> ### 5. Introduce references to appendixes in the main text
>
> The main text now contains explicit references to:  ablation studies appendix, FactScore-STEM-Geo construction appendix, and all appendix tables and figures are referenced from the sections that discuss the corresponding results.
>
> ### 6. Mention FactScore-STEM-Geo in the abstract and introduction
>
> We have added explicit mentions of FactScore-STEM-Geo in both the abstract ("We also introduce FactScore-STEM-Geo, a new 400-question long-form QA dataset spanning four categories across STEM and Geography") and in the introduction. We thank the reviewer for highlighting this contribution.

---

> > ### Comment · Reviewer_eJQG · 2026-05-07
> > **Response to Authors**
> >
> > Thank you for the updates and clarifications. All my comments were fully addressed.
> >
> > As a minor note, it would be nice if Appendix B had descriptions (similarly to Appendix A) about what experiment the supplemental materials are supplemental to and what they demonstrate.

---

> > > ### Author Response · Authors · 2026-05-12
> > > **Appendix B descriptions added**
> > >
> > > Thank you for the positive assessment and for confirming that all comments were fully addressed. We appreciate the suggestion regarding Appendix B. We have added brief descriptions to each subsection of Appendix B clarifying what each set of supplemental results demonstrates and linking back to the corresponding main-text sections. The updated PDF has been uploaded.

---

### Review · Reviewer_DG9P · 2026-04-19

**Summary Of Contributions:**

This paper presents a unified taxonomy for claim-level uncertainty quantification (UQ) in long-form LLM outputs, organized around a three-stage pipeline: response decomposition, unit-level scoring, and response-level aggregation. Within this framework, the authors formalize four families of black-box, consistency-based scorers, generalizing and extending prior methods such as LUQ, graph-based UQ, and long-form semantic entropy. The authors evaluate these scorers on four LLMs (Gemini-2.5-Flash/Pro, GPT-4o/4o-mini) using the FactScore-Bio dataset and a newly introduced FactScore-STEM-Geo dataset, which spans QA pairs for Chemistry, Physics/Math, Biology/Anatomy, and Geography. They assess unit-level classification, calibration, response-level correlation with factuality, and the effectiveness of uncertainty-aware decoding (UAD). They report that: (1) claim-response entailment performs as well or better than claim-level scorers while being substantially cheaper; (2) claim-level scoring outperforms sentence-level scoring in hallucination detection; and (3) UAD with claim-level filtering yields large factuality gains.

The main strengths of this work are:

- The three-stage taxonomy is clearly presented and provides a useful mental model for situating existing UQ methods. The formal notation in Section 3.1 allows existing approaches to be seen as products of specific design trade-offs within the same pipeline.

- The empirical evaluation is broad, spanning two datasets, four modern LLMs, two decomposition granularities, six semantic consistency functions, and four scorer families. The inclusion of verbalized confidence and short-form black-box baselines such as BERTScore, cosine similarity, non-contradiction and semantic negentropy provides additional context to the results.

- The new dataset FactScore-STEM-Geo complements the biography-centric focus of FactScore-Bio with domains that test methods' generalization to different types of factual content.

- The analysis in Section 5 is valuable to get a sense of which methods perform best under resource constraints, an aspect rarely reported systematically in the UQ literature. Moreover, the ablation study on sample size (Appendix A) provides useful empirical guidance (m=5 is typically sufficient), given that sampling dominates computational costs in UQ methods.

- The UAD experiments in Section 4.3 showcase a practical application of such methods to improve factual precision, beyond passive hallucination detection.

The weaknesses of this work are:

- The novelty of individual scorer extensions is modest, with most extensions representing a simple swap of consistency functions. The fact that this work's contribution lies mainly in a systematic taxonomy and comparison rather than in new methodological insights could be made more explicit in light of this.

- Response-level aggregation is underdeveloped relative to decomposition and unit-level scoring. Only simple averaging and UAD are considered, leaving the third stage of the three-stage pipeline substantially less explored than the other two.

- The response-level correlation analysis (Tables 3 and 10) is reported only for FactScore-Bio, weakening the generalization claims made in the Discussion section.

- The poor performance of claim-QA scorers, attributed in a footnote to atomic claims being poorly suited for question-inversion (Footnote 12), is a rather important methodological finding, and would probably deserve more than a footnote in light of reported results showing stronger factoid-level decomposition [3].

- Gemini-2.5-Flash is used both as a subject LLM being evaluated and as the grader/decomposer/question-generator, creating a potential source of self-preference bias that is acknowledged only obliquely. Consider using a different model for grading, and comparing to the original Gemini results to highlight (in)consistencies.

[1] Zhang et al. 2024. LUQ: Long-text Uncertainty Quantification for LLMs. https://aclanthology.org/2024.emnlp-main.299.pdf
[2] Jiang et al. 2024. Graph-based Uncertainty Metrics for Long-form Language Model Outputs. https://dl.acm.org/doi/10.5555/3737916.3738954
[3] Farquhar et al. 2024. Detecting hallucinations in large language models using semantic entropy. https://www.nature.com/articles/s41586-024-07421-0

**Audience:**

Yes

**Audience Explanation:**

Uncertainty quantification for LLMs is an active area in LLM research, and UQ for long-form outputs is very relevant given the widespread use of LLMs for drafting, summarization, retrieval-augmented generation, and agentic workflows. The taxonomy provides a useful shared vocabulary to make future work in this area easier to situate. As the authors note, prior work compared subsets of methods pairwise, and the broader comparison in this work can hence provide a valuable reference point for future papers. The practical recommendations are actionable and likely to influence downstream use of fine-grained UQ in applied settings, and suggest that UQ can move from a diagnostic tool to a generation-improvement mechanism, with implications for the controllable long-form generation.

Finally, the newly introduced FactScore-STEM-Geo dataset is a useful complement to existing benchmarks and expands the diversity of domains covered in long-form factuality evaluation. The promised open-source toolkit can also lower the bar for empirical comparison of future methods within the same framework.

**Broader Impact Concerns:**

The paper does not include a Broader Impact Statement. Given the topic, I believe a brief statement would be appropriate. The following concerns are worth noting:

- UAD produces responses that are both more accurate and more confident-seeming, which could increase over-reliance on outputs that still contain sneaky residual errors. This is relevant because the filtering step removes low-confidence claims rather than flagging them, so downstream users may lose visibility into the model's uncertainty. A brief discussion of how to communicate residual uncertainty to end users and how UAD interacts with this goal would be appropriate.

- The deployment of UQ methods without proper validation carries real risk, in particular in scientific domain such as those covered by FactScore-STEM-Geo. Noting this limitation and any implications for deployment in safety-sensitive domains would be appropriate.

**Claims And Evidence:**

Yes

**Claims Explanation:**

The three headline claims are supported (with some caveats) by the experimental results.

The first claim, "claim-response entailment is better or on par with more complex claim-level scorers," is supported by Table 2, Figure 5, and Table 3. Claim-response entailment achieves the highest AUPRC in all 8 LLM-dataset combinations, is within 0.01 AUROC of the top scorer in 6 of 8, and achieves the strongest or near-strongest response-level Pearson correlations, while requiring roughly 1/3 of the generations used by graph-based scorers (Table 4).

The second claim, "claim-level scoring generally yields better results than sentence-level scoring," is supported by the AUROC gap visible in Table 2 (top claim-level AUROC 0.67-0.80 vs. top sentence-level AUROC 0.57-0.70) and the response-level correlations in Table 3. Provided the gap between these is not uniform, sentence-level scoring retains value when claim decomposition is infeasible.

The third claim regarding UAD effectiveness is strongly supported by Figure 6, which consistently shows substantial accuracy gains across all 8 LLM-dataset combinations. The +0.18 accuracy at moderate filtering is a considerable effect given the literature on LLM factuality evaluation.

**Requested Changes:**

I have marked changes I consider critical to acceptance (C) versus those that would strengthen the work but are not strictly required (S).

(C) Report response-level correlation results (Tables 3 and 10) for FactScore-STEM-Geo as well, to match the scope of the unit-level analysis.

(C) Clarify the novelty of individual scorer variants. A table mapping each scorer to its prior-work origin (exact method / generalization of / new) would help readers calibrate expectations. The acknowledgment in Section 3.3.1 that sentence- and claim-level scoring with contrasted entailment are equivalent to LUQ and LUQ-atomic can be used as a model for Sections 3.3.2 to 3.3.4.

(C) Address the potential self-preference bias from using Gemini-2.5-Flash as both a subject LLM and as grader/decomposer/question-generator. Even a small-scale robustness check with a different grader (e.g., GPT-4o) on a subset of the data would substantially strengthen the empirical claims.

(S) Expand the analysis of claim-QA's poor performance beyond the footnote. Evaluating factoid-level decomposition (as in [3]) as an intermediate granularity between atomic claims and sentences would test whether the issue is with question-inversion in general or with the specific atomic-claim decomposition used.

(S) Include at least one white-box baseline (e.g., average token negative log-probability, response-level perplexity). While the Limitations section notes this omission, even a single method in this category would help readers assess the difference between black-box methods and white-box probability aggregations.

(S) Include additional aggregation strategies beyond simple averaging and UAD in Section 3.4, for example, minimum (worst-unit), geometric mean, or rank-weighted aggregation. The three-stage framework motivates this but the aggregation stage receives considerably less attention than decomposition and unit scoring.

(S) Discuss the potential selection bias in FactScore-STEM-Geo arising from choosing the 100 longest Wikipedia articles per category. This likely biases the evaluation toward well-documented entities; the results may not generalize to less-documented entities, where the risk of hallucinations is arguably higher.

---

> ### Author Response · Authors · 2026-05-01
> **Revisions made with all suggestions added**
>
> We thank the reviewer for their thorough and constructive feedback.
>
> ### Response-level correlation for FactScore-STEM-Geo (C)
> Response-level correlation for FactScore-STEM-Geo (C). We have added response-level correlation results for FactScore-STEM-Geo in Tables 16-17 (Appendix). Response-level correlations are generally weaker than on FactScore-Bio (top correlations 0.37–0.62 vs. 0.60–0.74), consistent with the unit-level evaluation. The key finding that claim-level scoring outperforms sentence-level scoring continues to hold. We observe some shifts in relative scorer rankings across datasets, which may reflect compounding of noisier unit-level scores under aggregation.
>
> ### Clarifying novelty of scorer variants (C)
> We have added Table 2, which maps each scorer to its prior-work origin (exact method, generalization, or new contribution). We appreciate this suggestion and believe it strengthens the paper by clarifying our exact contributions.
>
> ### Self-preference bias (C)
> We conducted a robustness check using GPT-4o as an alternative grader on a stratified sample of the data (n=400 per generator-granularity combination). Results are reported in Appendix B. We observe substantial to almost perfect inter-grader agreement (Cohen's κ = 0.63–0.84), with claim-level grading showing higher agreement than sentence-level grading. Crucially, agreement levels are consistent regardless of whether the generator LLM matches the original grader, suggesting that self-preference bias does not substantially affect our results.
>
> ### Claim-QA's poor performance (S)
> We have expanded the discussion of claim-QA's limitations in Section 5, moving beyond the footnote to a dedicated paragraph. The core issue is that atomic claims create an inherent tension with QA-based verification: questions that preserve the claim's meaning often admit multiple correct answers, while questions yielding unique answers tend to encode the answer within the question itself. This helps explain why sentence-QA scorers outperform claim-QA methods despite sentence-level classification being an inherently harder task. We agree that evaluating factoid-level decomposition as an intermediate granularity is a promising direction, and we note this as a suggestion for future work in Section 7 given the scope of the current evaluation.
>
> ### White‑Box Baselines (S)
> We have updated the manuscript (Table 2 and Table 3) to include white‑box baselines (average token log‑probabilities). These metrics are provided for the four models included in our original experimental design (Gemini‑2.5 and GPT‑4o families), for which internal confidence scores were captured incidentally during the primary generation phase. White‑box scores were not part of the paper’s original scope, but because we happened to record them for these models, we have now added them as baselines.
>
> **Regarding the omission of white‑box scores for Llama‑4‑Maverick‑17B:**
>
> This model was added during the revision process to satisfy a specific request from Reviewer 1 to expand the experimental scope. Because it was not part of the original data‑logging pipeline, we did not capture log‑probabilities for it at that time. The Together AI serverless endpoint used for this model has since been retired, and alternative providers (e.g., OpenRouter, SambaNova) do not currently expose log‑probability access for this specific 17B architecture. Running the model locally to extract log‑probabilities is infeasible given our available hardware resources. Consequently, we are not able to feasibly obtain white‑box scores  for this model.
>
>
> ### Additional aggregation strategies (S)
> We have evaluated minimum, geometric mean, and rank-weighted aggregation. While averaging is our primary aggregation method, we additionally consider these alternatives in Sections 3.4 and 4.4. Simple averaging consistently matched or outperformed alternatives; minimum aggregation performed notably worse, suggesting isolated low-confidence claims are poor indicators of overall response factuality. In our discussion, we note that minimum aggregation may be preferred in high-risk applications where flagging any potentially unsupported claim matters more than estimating average factuality.
>
> ### Selection bias in FactScore-STEM-Geo (S)
> We have added a discussion of this limitation in Section 7. We agree that selecting the longest Wikipedia articles potentially biases toward well-documented entities, and we note that generalization to less-documented entities warrants further investigation.
>
> ### Broader Impact Statement
> We have added a Broader Impact discussion in Section 5 addressing these concerns. We discuss how UAD's removal of low-confidence claims may reduce user visibility into model uncertainty, and note the risks of deploying UQ methods without proper validation in safety-sensitive domains.

---

### Review · Reviewer_9sfN · 2026-04-29

**Summary Of Contributions:**

This paper proposes a framework for Uncertainty Quantification of long-form black-box LLM responses, which generalizes several existing works. The framework consists of 3 main components: Decomposition into atomic claims, scoring individual claims, and aggregating claim scores.

For decomposition, the paper evaluates LLM-based decomposition into phrase-level claims, and sentence splitting. To score individual claims, the paper evaluates
* Two approaches (Unit-Response Scores, Matched-Unit Scores) where claims are compared to alternative responses (or claims contained therein), for example by NLI, cosine embedding similarity, or semantic similarity such as BertScore;
* An approach (Unit-QA Score) where claims are scored by (1) generating a question for which the claim is the answer; (2) sampling responses to the generated question from an LLM; (3) comparing the similarity of generated responses to the original claim and averaging ;
* A graph-based approach (Graph-Based Score) where claims from the original and alternative responses are combined and form a set of nodes. An edge exists between two nodes that entail each other.

In all approaches, the paper compares different design choices, for example the comparison score (NLI, cosine, ...) in Unit-Response Scores or the graph metric in Graph-Based Scores.

Finally, the paper also evaluates the effectiveness of claim scoring by reconstructing responses after filtering all low-confidence scores and observing performance improvements.

**Audience:**

Yes

**Audience Explanation:**

Uncertainty Quantification and Hallucination Detection are important topics that are of interest to a wide audience in LLM research.

**Broader Impact Concerns:**

There are no specific ethical implications or concerns that need to be discussed.

**Claims And Evidence:**

Yes

**Claims Explanation:**

Overall, the proposed taxonomy is convincing and the paper evaluates a significant number of variations, which directly support the claims mentioned in the abstract.

All notable limitations of the experimental setup are explicitly mentioned in the paper:
* No evaluation of matched-claim scores, the paper mentions too high computational complexity
* No ablation of NLI model, claim decomposition and merging LLM, and similar model choices

Nonetheless, the evaluation is sufficiently comprehensive to yield useful and likely generalizable insights. The experiment on uncertainty guided generation also serves as a useful confirmation of the strengths of the evaluated methods.

**Requested Changes:**

While I agree that the main experiments are sufficient to support the claims in the paper, I think the following additions are neccessary:

**(Q1)** The paper should explicitly define metrics. This mainly concerns the FactScore which should be briefly described for the paper to be self contained. Furthermore, I could not find any explicit definition of the "LLM Accuracy" metric or method, which is for example used in Tab. 2.

**(Q2)** The paper should include an evaluation of the LLM-based claim decomposition. This could be done on a subset of the benchmarks or on a related dataset which has claim span labels. Since the claim decomposition forms the basis of all other methods, it is very important to understand how well it works.

**(Q3)** The paper should also include an analysis how well score thresholds generalize between tasks. For example, metrics like AUROC can be high for different tasks separately, but have distinct scales. In practice, however, a system for hallucination detection will rely on a fixed threshold which has to be fixed using a specific dataset, so generalization is important.

---

> ### Author Response · Authors · 2026-05-01
> **Revisions made with all suggestions added**
>
> We thank the reviewer for these helpful and constructive suggestions.
>
> ### Response to Q1
>
> We have added explicit definitions of the requested metrics in the revised paper.
>
> **LLM Accuracy / FactScore grading** refers to the proportion of units (claims or sentences) in a given LLM's response that are graded as factually correct under the FactScore protocol. Specifically, each response is decomposed into granular units, and each unit is compared against the corresponding Wikipedia article text by a grader LLM (Gemini-2.5-Flash in our experiments). The grader is prompted with the Wikipedia context and the unit, and returns a binary supported/not-supported judgment (see Listing 1-2 in Appendix D for prompt templates). LLM Accuracy is then the fraction of units receiving a "supported" judgment, computed per LLM and reported in Table 3. Following [Jiang et al., 2024](https://arxiv.org/abs/2410.20783), for claim-level evaluation, we further classify claims as objective or subjective and retain only objective claims, since subjective ones cannot be definitively verified against the reference text. We have clarified this definition in Section 4.1 of the revised manuscript and ensured consistent terminology throughout the paper.
>
>
> ### Response to Q2
>
> We have added a manual evaluation of the LLM-based claim decomposition to assess both faithfulness and coverage (Appendix A.2).
>
> **Evaluation Setup:** Two authors jointly evaluated 410 claims across 15 responses (3 randomly sampled per dataset for FactScore-Bio and each of the four FactScore-STEM-Geo categories). Each claim was assessed for (1) faithfulness to the source response, (2) standalone interpretability per the decomposition instructions, and (3) coverage of the original content.
>
> **Results:**
>
> | Criterion | Result | Rate |
> |-----------|--------|------|
> | Faithfulness | 409/410 claims factually correct | 99.8% |
> | Standalone quality | 397/410 claims interpretable without context | 96.8% |
> | Recall | 3 claims missing across all responses | ~99.3% |
>
> Only 1/410 claims (0.2%) contained an objective factual error relative to the source response. An additional 13/410 claims (3.2%) were factually correct but required surrounding context to interpret, violating the standalone decomposition requirement—these represent decomposition artifacts rather than faithfulness errors.
>
> These results demonstrate that the claim decomposition is highly reliable, with near-perfect faithfulness and recall. The small proportion of context-dependent claims does not affect downstream verification, as the claims remain factually accurate.
>
> ### Response to Q3
>
> We have added a new analysis (Appendix A.3) investigating threshold generalization across both datasets and scorers.
>
> **Cross-dataset transfer:** We find that thresholds generalize well across datasets when using the same scorer. For claim-level scorers, transferring a threshold tuned on one dataset to another incurs minimal performance loss (gaps up to 0.028 F1). Claim-QA noncontradiction shows near-perfect transfer with essentially zero gap on both datasets. Sentence-level scorers exhibit slightly larger gaps (up to 0.049), with QA-cosine similarity showing the best transfer (gaps of 0.007 and 0.000).
>
> **Cross-scorer transfer:** In contrast, transferring thresholds between different scorers performs substantially worse. While claim-level scorers show relatively robust transfer, sentence-level scorers exhibit severe degradation when transferring from QA-cosine similarity to entailment-based scorers (F1 dropping from 0.605 to as low as 0.312). This is expected since different scorers produce scores with different distributions.
>
> These results suggest that practitioners can tune a threshold on a labeled sample from one domain and apply it to new domains with minimal performance loss, as long as the same scorer is used. However, switching to a different scorer requires re-tuning the threshold. This has practical implications for deployment, as the expensive labeling effort transfers across domains but not across scoring methods.

---

### Decision · Action_Editor_vSbb · 2026-05-23

**Recommendation:** Accept as is

**Audience:**

Yes

**Audience Explanation:**

Yes, language modeling, hallucination detection, factuality and uncertainty quantification research communities will be interested in these findings.

**Claims And Evidence:**

Yes

**Claims Explanation:**

The paper studies uncertainty and hallucination in long form responses of LLMs, and the authors rigorously support all claims with experiments.

---

> ### Author Response · Authors · 2026-05-27
> **Camera-ready version submitted**
>
> Dear AE,
>
> Thank you very much for handling our paper and for your time and effort throughout the process. We have uploaded the camera-ready version and included the link to our code repository.
>
> Please let us know if there are any further questions.
>
> Sincerely,
> Paper 7590 Authors

---

> > ### Comment · Action_Editor_vSbb · 2026-06-22
> > **Camera Ready Verification**
> >
> > Dear authors,
> >
> > Congratulations on the acceptance of your paper!
> > I apologize for the delay in the process.
> >
> > Best,
> >
> > AE